# SEQUENCE ANALYSIS USING THE BÉZIER CURVE

## ABSTRACT

The analysis of sequences (e.g., protein, DNA, and SMILES string) is essential for disease diagnosis, biomaterial engineering, genetic engineering, and drug discovery domains. Conventional analytical methods focus on transforming sequences into numerical representations for applying machine learning/deep learning-based sequence characterization. However, their efficacy is constrained by the intrinsic nature of deep learning (DL) models, which tend to exhibit suboptimal performance when applied to tabular data. An alternative group of methodologies endeavors to convert biological sequences into image forms by applying the concept of Chaos Game Representation (CGR). However, a noteworthy drawback of these methods lies in their tendency to map individual elements of the sequence onto a relatively small subset of designated pixels within the generated image. The resulting sparse image representation may not adequately encapsulate the comprehensive sequence information, potentially resulting in suboptimal predictions. In this study, we introduce a novel approach to transform sequences into images using the Bézier curve concept for element mapping. Mapping the elements onto a curve enhances the sequence information representation in the respective images, hence yielding better DL-based classification performance. We employed three distinct protein sequence datasets to validate our system by doing three different classification tasks, and the results illustrate that our Bézier curve method is able to achieve good performance for all the tasks. For instance, it has shown tremendous improvement for a protein subcellular location prediction task over the baseline methods, such as improved accuracy by 39.4% as compared to the FCGR baseline technique using a 2-layer CNN classifier. Moreover, for Coronavirus host classification, our Bézier method has achieved 5.3% more AUC ROC score than the FCGR using a 3-layer CNN classifier.

## 1 INTRODUCTION

Sequence analysis, especially protein sequence analysis (Whisstock & Lesk, 2003; Hirokawa et al., 1998), serves as a foundational undertaking within the field of bioinformatics, possessing a broad spectrum of applications encompassing drug exploration, ailment detection, and tailored medical interventions. The comprehension of attributes, functionalities, configurations, and evolutionary patterns inherent to biological sequences holds paramount significance for elucidating biological mechanisms and formulating effective therapeutic approaches (Rognan, 2007).

Traditional phylogenetic approaches (Hadfield et al., 2018; Minh et al., 2020) for the analysis of biological sequences are no longer effective due to the availability of large sequence data, as these methods are not scalable due to being computationally very expensive. They also require extensive domain knowledge, and incomplete knowledge easily hinders the results. Numerous feature-engineering-based works exist to encode sequences into numerical form to perform machine learning (ML)/Deep learning (DL)-based analysis, as ML/DL models are well-known to tackle large datasets efficiently. For example, OHE (Kuzmin et al., 2020) builds binary vectors against the sequences. However, it is alignment-based, and sequence alignment is an expensive process. The generated vectors by OHE are also very sparse and highly dimensional. Another set of approaches (Ali & Patterson, 2021; Ma et al., 2020) follows the $k$-mers concept to obtain feature embeddings. But they also undergo sparsity challenges and are usually computationally expensive. Moreover, some approaches (Shen et al., 2018; Xie et al., 2016) utilize a neural network to extract feature embeddings

from the sequences to perform analysis. However, they usually require large training data to achieve optimal performance, and acquiring more data is usually an expensive procedure for medical data.

An alternative approach for biological sequence analysis entails the transformation of sequences into image representations. This adaptation facilitates the utilization of sophisticated DL vision models to address sequence analysis objectives, as DL models are very popular in achieving state-of-the-art performance for image classification. FCGR (Löchel et al., 2020), RandomCGR (Murad et al.), and protein secondary structure prediction (Zervou et al., 2021) are some of the methods that fall under this category. They are based on the concept of CGR(Chaos Game Representation) (Jeffrey, 1990). Such mappings are between amino acids and specific pixels of the generated images, which can result in suboptimal representation due to capturing the information in a sparse way about amino acids/nucleotides of a sequence in its respective constructed image.

Therefore, in this work, we propose a method based on the Bézier curve (Han et al., 2008) to translate biological sequences into images to enable the application of DL models on them. Bézier curve (Han et al., 2008) is a smooth and continuous parametric curve that is defined by a set of discrete control points. It is widely used to draw shapes, especially in computer graphics and animation. It has been used in the representation learning domain previously but mainly focusing on extracting numerical features, such as in (Hug et al., 2020) which does n-step sequence prediction based on the Bézier curve, (Liu et al., 2021) proposed end-to-end text spotting using the Bézier curve, (Qiao et al., 2023) does map construction, etc. However, we aim to utilize the Bézier curve to formulate an efficient mechanism for transforming biological sequences into images by effectively mapping the components of a sequence onto a curve. Each component, or character (an amino acid, nucleotide, etc.) of a sequence is represented by multiple lines on the curve which enable more information to be captured in the respective image, hence producing a better representation. The goal of using Bezier curves is to create a visualization that aids in the analysis of protein sequences. This visualization can allow researchers to explore patterns and trends that might provide insights into protein structure and function.

Our contributions in this work are as follows,

1. We present a novel approach for converting biological sequences into images utilizing the Bézier function. By harnessing the capabilities of the Bézier curve in conjunction with deep learning analytical models, we can foster a more profound comprehension of these sequences. This innovative technique holds promise for advancing our understanding of biological data and enabling more robust analysis and insights.
2. Using three distinct protein datasets (protein subcellular dataset, Coronavirus host dataset, ACP dataset) for validating our proposed technique, we show that our method is able to achieve high performance in terms of predictive performance for various classification tasks.

The rest of the paper is organized as follows: Section 2 talks about the literature review, Section 3 discusses the proposed approach in detail, Section 4 highlights the experimental setup details of our work, Section 5 discusses the results obtained from the experiments, and Section 6 concludes the paper.

## 2 LITERATURE REVIEW

Biological sequence analysis is an active research area in the domain of bioinformatics. Numerous works exist to tackle biological sequences, and most of them aim to map sequences into machine-readable form to perform further ML/DL-based analysis on them. For instance, OHE (Kuzmin et al., 2020) constructs binary vectors to represent the sequences, but these vectors are very sparse and suffer from the curse of dimensionality challenge. Likewise, Spike2Vec (Ali & Patterson, 2021) & PWkmer (Ma et al., 2020) design feature embeddings based on the $k$-mers of the sequences. However, they also undergo the sparsity issue, and computation of $k$-mers is usually an expensive process, especially for long sequences. Moreover, some approaches (Shen et al., 2018; Xie et al., 2016) employ a neural network to obtain the numerical embeddings of the sequences, but their large training data requirement to attain optimal performance is an expensive requirement. Furthermore, a set of works (Protein Bert (Brandes et al., 2022), Seqvec (Heinzinger et al., 2019), UDSMProt (Strodthoff et al., 2020)) follows the utilization of pre-trained models for extracting features from the protein

sequences to assist the classification tasks. However, these mechanisms are computationally very costly. Several kernel matrix-based works (Farhan et al., 2017; Ali et al., 2022) are put forward to deal with protein sequence classification. These methods build a symmetric kernel matrix to represent the sequences by capturing the similarity between them, and this matrix is further utilized as input to the classification tasks. But the kernel matrix is usually of high dimensions, and loading it is memory inefficient. An alternative set of techniques transforms the sequences into images, particularly for enabling the application of sophistical DL analytical models in the domain of bio-sequence analysis. These methodologies (Murad et al., 2023; Zervou et al., 2021; Murad et al.; Löchel et al., 2020) are usually built upon the concept of CGR (Jeffrey, 1990). They follow an iterative mechanism to construct the images. However, these methods map the components (amino acids/nucleotides) of a sequence to specific pixels in the corresponding generated image, while our method maps them onto a Bézier curve, resulting in more intuitive and easy-to-interpret visualization.

## 3 PROPOSED APPROACH

This section discusses the details of our proposed method, which converts protein sequences into images following the concept of the Bézier curve to enable the application of sophisticated DL models on the sequence classification tasks.

The general formula (Baydas & Karakas, 2019) of the Bézier curve is

$BZ(t) = \Sigma_{i=0}^{n} \binom{n}{i} t^i (1-t)^{n-i} P_i$ where $0 \leq t \leq 1$, $P_i$ are known as control points and are elements of $\mathbb{R}^k$, and $k \leq n$.

To construct the protein images, we employ a Bézier curve with $n = 3$ and $k = 2$. As images consist of x and y coordinates, therefore $k = 2$ is used. The formulas to determine the coordinates for representing an amino acid in the respective generated image are,

$x = (1-t)^3 \cdot P_{0_x} + 3 \cdot (1-t)^2 \cdot t \cdot P_{1_x} + 3 \cdot (1-t) \cdot t^2 \cdot P_{2_x} + t^3 \cdot P_{3_x}$ (1)

$y = (1-t)^3 \cdot P_{0_y} + 3 \cdot (1-t)^2 \cdot t \cdot P_{1_y} + 3 \cdot (1-t) \cdot t^2 \cdot P_{2_y} + t^3 \cdot P_{3_y}$ (2)

where, $(P_{0_x}, P_{0_y})$, $(P_{1_x}, P_{1_y})$, $(P_{2_x}, P_{2_y})$, & $(P_{3_x}, P_{3_y})$ denote the x & y coordinates of the four distinct control points respectively.

The algorithm and workflow of creating Bézier-based images are illustrated in Algorithm 1 and Figure 1 respectively. We can observe that given a sequence and number of parameters $m$ as input, the algorithm and workflow yield an image as output. Note that $m$ indicates the parameter $t$ shown in the above equations. The process starts by computing the control points by considering the unique amino acids of the given sequence and their respective ASCII values (numerical), as depicted in steps 4-6 of the algorithm and step (b) of the workflow. A control point is made of a pair of numerical values representing the x and y coordinates, where x is assigned the index of the first occurrence of the respective unique amino acid and y holds its ASCII value. Moreover, $m$ linearly spaced random pairs belonging to [0,1] are generated as parameters (mentioned in step 9 and step (c) of the algorithm and workflow respectively). Note that we used $m = 200$ for our experiments. Then the deviation pair points are generated for every amino acid of the sequence (as exhibited in step 15 of the algorithm and step (d) of the workflow). We utilized 3 deviation pairs to conduct our experiments. After that, modified pair points are obtained by adding the deviation pairs to the corresponding amino acid's control point pair respectively, as shown in step 16 of the algorithm and step (e) of the workflow. Then the Bézier pair points are extracted from the Bézier function by employing equation 1 and equation 2 (as presented in step 19 and step (f) of the algorithm and workflow respectively). Finally, the Bézier pairs are used as x and y coordinates to plot the image (as shown in step 23 and step (g) of the algorithm and workflow respectively). Note that, we get multiple Bézier pairs depending on the value of $m$ and we plot all the pairs in the created image to represent the respective amino acid in the image.

As Bézier curves are known for their ability to smoothly interpolate control points, using them to connect control points for representing amino acids ensures a visually smooth transition between points, making the visualization more intuitive and easy to interpret. Moreover, introducing randomness to the control points by adding deviations results in controlled CGR. While the approach deviates from traditional CGR, it helps reveal patterns that might not be apparent in regular CGR due to the scattering of control points. This randomness mimics the inherent variability and noise

---

**Algorithm 1** Bézier Curve Based Image Generation

---

1: **Input:** Sequence $seq$, No. of Parameters $m$
2: **Output:** Image $img$
3: conPoint = {}                                          ▷ dictionary for control points
4: **for** $i, aa \in seq$ **do**:                        ▷ every unique amino acid aa in seq
5:     conPoint[aa] = $[i, ASCII(aa)]$      ▷ assign control point the index i and ASCII of aa
6: **end for**
7: $xCord$ = []                                           ▷ list for x coordinates
8: $yCord$ = []                                           ▷ list for y coordinates
9: $t\_Val$ = Get $m$ pairs $\in [0, 1]$                  ▷ list of m pairs of parameters
10: $ite$ = 3                              ▷ no of deviations pair points. It can have any value.
11: **for** $a \in seq$ : **do**                          ▷ every amino acid a in seq
12:     org_point = conPoint[a]                           ▷ control point of $a$
13:     points = [org_point]
14:     **for** $i \in (ite)$ : **do**
15:         dev = Get_Random_Pair                         ▷ get a random pair
16:         mod_point = org_point + dev                   ▷ get a modified control point
17:         points.append(mod_point)
18:     **end for**
19:     curve_point = Get_Bezier_Point(points, $t\_Val$)   ▷ get bezier curve points from bezier func
20:     $xCord$ = curve_point[:0]                         ▷ get x coords of curve
21:     $yCord$ = curve_point[:1]                         ▷ get y coords of curve
22: **end for**
23: $img$ = plot($xCord, yCord$)                          ▷ get image by plotting x & y coords
24: return($img$)

---

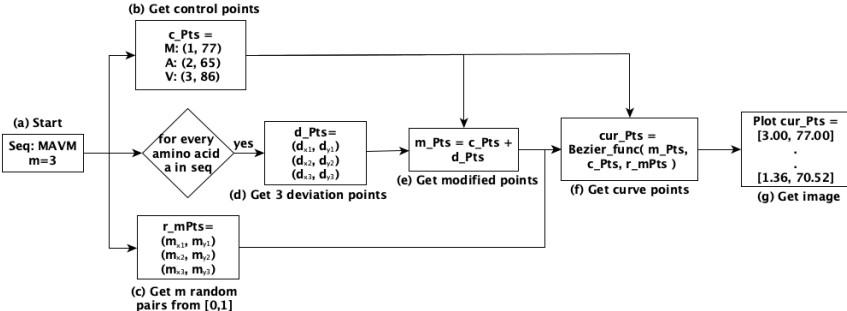

Figure 1: The workflow of our system to create an image from a given sequence and a number of parameters $m$. We have used "MAVM" as an input sequence here. Note that the $cur\_Pts$ consists of a set of values for x coordinates and y coordinates.

present in biological sequences. It can be justified as an attempt to capture the inherent variability in protein sequences that can arise due to mutations, structural differences, or experimental variations.

## 4    EXPERIMENTAL EVALUATION

This section discusses the details of the experimental setup used to perform the experiments. It highlights the datasets, baseline methods, and classification models. All experiments are carried out on a server having Intel(R) Xeon(R) CPU E7-4850 v4 @ 2.40GHz with Ubuntu 64-bit OS (16.04.7 LTS Xenial Xerus) having 3023 GB memory. We employed Python for implementation and the code is available online for reproducability [1].

---

[1] Available in the published version

## 4.1 DATA STATISTICS

We have used 3 distinct protein sequence datasets, a nucleotide-based dataset, a musical dataset, and a SMILES string dataset to evaluate our proposed system. The reason to use such diversified datasets is to show the generalizability of our method for any type of sequence. Each dataset is summarized in Table 1. Further details of the datasets are given in Appendix C.

Table 1: The summary of all the datasets used to perform the evaluation of our method.

| Dataset | Description |
| --- | --- |
| Protein Subcellular Localization | It has $5959$ unaligned protein sequences distributed among $11$ unique subcellular locations. The associated subcellular location is predicted for a given protein sequence as input. |
| Coronavirus Host | The unaligned spike protein sequences from various clades of the Coronaviridae family are collected to form this dataset. It contains $5558$ spike sequences distributed among $22$ unique hosts. |
| Anticancer Peptides (ACPs) | It consists of $949$ unaligned peptide-protein sequences along with their respective anticancer activity on the breast cancer cell lines distributed among the $4$ unique target labels. |
| Human DNA (Human DNA) | It consists of $2,000$ unaligned Human DNA nucleotide sequences which are distributed among seven unique gene families. These gene families are used as labels for classification. The gene families are G Protein Coupled, Tyrosine Kinase, Tyrosine Phosphatase, Synthetase, Synthase, Ion Channel, and Transcription Factor containing $215$, $299$, $127$, $347$, $319$, $94$, & $599$ instances respectively. |
| SMILES String (Shamay et al., 2018) | It has $6,568$ SMILES strings distributed among ten unique drug subtypes extracted from the DrugBank dataset. We employ the drug subtypes as a label for doing classification. The drug subtypes are Barbiturate [EPC], Amide Local Anesthetic [EPC], Non-Standardized Plant Allergenic Extract [EPC], Sulfonylurea [EPC], Corticosteroid [EPC], Nonsteroidal Anti-inflammatory Drug [EPC], Nucleoside Metabolic Inhibitor [EPC], Nitroimidazole Antimicrobial [EPC], Muscle Relaxant [EPC], and Others with $54$, $53$, $30$, $17$, $16$, $15$, $11$, $10$, $10$, & $6352$ instances respectively. |
| Music Genre (Li et al., 2003) | This data has $1,000$ audio sequences belonging to $10$ unique music genres, where each genre contains $100$ sequences. We perform music genre classification tasks using this dataset. The genres are Blues, Classical, Country, Disco, Hiphop, Jazz, Metal, Pop, Reggae, and Rock. |

## 4.2 BASELINE MODELS

We compared the performance of our proposed method with various baselines. These baselines are categorized into three groups: feature-engineering-based baselines, kernel-based baseline, and image-based baselines. The feature-engineering-based baselines (OHE (Kuzmin et al., 2020)), WD-GRL (Shen et al., 2018)) consist of methods that map the bio-sequences into numerical vectors to enable the application of ML/DL models on them. In the kernel-based baseline (String kernel (Ali et al., 2022; Farhan et al., 2017)), the goal is to design a kernel matrix and then use kernel PCA to get the final embeddings, which can then be used as input to classical ML models, like SVM, Naive Bayes (NB), Multi-Layer Perceptron (MLP), K-Nearest Neighbors (KNN), Random Forest (RF), Logistic Regression (LR), and Decision Tree (DT), to perform sequence classification. The image-based baselines (FCGR (Löchel et al., 2020), RandomCGR (Murad et al.), Spike2CGR (Murad et al., 2023)) aim to transform the bio-sequences into images to perform DL-based classification. The baseline methods used are summarized in Table 2 and their further details are mentioned in Appendix D.

## 4.3 CLASSIFICATION MODELS

In the realm of classification tasks, we have employed two distinct categories of classifiers: Image models and Tabular models. For both categories, the data follows $80-20\%$ split for train-test sets, and the train set is further divided into $70-30\%$ train-validation sets. These splits follow a stratified sampling strategy to keep the distribution the same as given in the original data.

Table 2: The summary of all the baseline methods which are used to perform the evaluation.

| Category | Method | Description |
|---|---|---|
| Feature Engineering based methods | OHE | It generates binary vector-based numerical embeddings of the sequences. |
| | WDGRL | It is an unsupervised approach that uses a neural network to extract numerical features from the sequences. |
| Kernel Method | String Kernel | Given a set of sequences as input, this method designs $n \times n$ kernel matrix that can be used with kernel classifiers or with kernel PCA to get feature vectors |
| Image based methods | FCGR | It maps the protein sequences into images by following the concept of CGR and constructs n-flakes-based images. |
| | RandomCGR | This method follows a random function for determining the coordinates of amino acids of protein sequences to create images. |
| | Spike2CGR | This technique combines CGR with minimizers and $k$-mers concepts to determine the images of given protein sequences. |

### 4.3.1 IMAGE MODELS

These models are used for image-based classification. We construct four custom convolutional neural networks (CNNs) classifiers with varying numbers of hidden layers to do the classification tasks. These models are referred to as 1-layer, 2-layer, 3-layer & 4-layer CNN classifiers, and they consist of 1, 2, 3, & 4 hidden block A modules respectively. A block A module contains a convolution layer followed by a ReLu activation function and a max-pool layer. These custom CNN networks are employed to investigate the impact of increasing the number of hidden layers on the final predictive performance. Moreover, a vision transformer model (ViT) is also used by us for performing the classification tasks. As ViT is known to utilize the power of transformer architecture, we want to see its impact on our bio-sequence datasets classifications. Furthermore, we also examine the consequences of using pre-trained vision models for classifying our datasets, and for that, we used pre-trained ResNet-50 (He et al., 2016), EfficientNet (Tan & Le, 2019), DenseNet (Iandola et al., 2014) and VGG19 (Simonyan & Zisserman, 2015) models. The image classifiers are summarized in Table 3, and further details about their respective architectures and hyperparameters can be viewed in Appendix E.1.1.

Table 3: The summary of all the image models used to perform the evaluation through image classification.

| Category | Model | Description |
|---|---|---|
| Custom CNN | 1-layer CNN | A custom CNN model with one hidden block A module (layer). |
| | 2-layer CNN | A custom CNN model with two hidden block A modules (layers). |
| | 3-layer CNN | A custom CNN model with three hidden block A modules (layers). |
| | 4-layer CNN | A custom CNN model with four hidden block A modules (layers). |
| Transformer | ViT | A vision transformer classifier following the architecture of the transformer to do image-based classification. |
| Pre-trained | VGG19 | The pre-trained VGG19 (Simonyan & Zisserman, 2015) is employed to do image-based classification. |
| | ResNet-50 | The pre-trained ResNet-50 (He et al., 2016) is employed to do image-based classification. |
| | EfficientNet | The pre-trained EfficientNet (Tan & Le, 2019) is employed to do image-based classification. |
| | DenseNet | The pre-trained DenseNet (Iandola et al., 2014) is employed to do image-based classification. |

### 4.3.2 TABULAR MODELS

These models aim to classify the numerical data. We have used two distinct DL tabular models in our experimentation, which are known as the 3-layer tab CNN model & the 4-layer tab CNN model. 3-layer tab CNN model consists of 3 hidden linear layers, while 4-layer tab CNN has 4 hidden linear layers. In each of the classifiers, the hidden layers are followed by a classification linear layer. The hyperparameters chosen by us after fine-tuning are 0.001 learning rate, ADAM optimizer, NLL loss function, and 10 training epochs. Moreover, the input vectors from WDGRL are of dimensions 10, as it transforms the data into low dimensional space. Furthermore, we employed some ML models (SVM, Naive Bayes (NB), Multi-Layer Perceptron (MLP), K-Nearest Neighbors (KNN), Random Forest (RF), Logistic Regression (LR), and Decision Tree (DT)) to classify the kernel-method-based feature embeddings.

# 5 RESULTS AND DISCUSSION

This section provides an extensive discussion of the classification results obtained by our proposed method and the baseline approaches for 3 distinct classification tasks using 3 different datasets respectively.

## 5.1 PROTEIN SUBCELLULAR DATASET'S PERFORMENCE

The classification results of the protein subcellular dataset via different evaluation metrics are mentioned in Table 4. We can observe that in the case of the custom CNN models, the performance stopped increasing after two layers. It could be because of the dataset being small in size which causes the gradient vanishing problem. Moreover, for the ViT model although the Bézier images have maximum performance as compared to the FCGR and RandomCGR images, however, the overall performance gained by the ViT model is less than the custom CNN models. A reason for this could be the dataset being small in size as ViT typically requires substantial training data to surpass CNN models. Additionally, in ViT a global attention mechanism is used which focuses on the entire image, but in the images generated by all three methods (FCGR, RandomCGR & Bézier) the pertinent information is concentrated in specific pixels, with the remaining areas being empty. Consequently, the global attention mechanism may not be as efficient for these images as a local operation-based CNN model, which is tailored to capture localized features efficiently. The feature-engineering-based methods are yielding very low performance as compared to our image-based methods (especially FCGR & Bézier) indicating that the image-based representation of bio-sequences is more effective in terms of classification performance over the tabular one. The pre-trained ResNet-50 classifier corresponding to the Bézier method has the optimal predictive performance for all the evaluation metrics. It shows that the ResNet-50 is able to generalize well to the Bézier generated images. It may be due to the architecture of ResNet (like skip connections) enabling the learning on our small dataset. Overall, the pre-trained models (ResNet, VGG19, & EfficientNet) are performing well for the Bézier based images, except the DensetNet model. A reason for DenseNet having very bad performance could be the dataset being small, as DenseNet typically requires large data to yield good performance. Furthermore, among the image-based methods, our Bézier method is tremendously outperforming the baselines for every evaluation metric corresponding to all the vision DL classifiers. This can be because the average length of sequences in the protein subcellular localization dataset is large and our technique uses the Bézier curve to map each amino acid, so a large number of amino acids results in more effective capturing of information about the sequences in their respective constructed images.

We have also added results of the Spike2CGR baseline method in Table 4 and we can observe that this method is underperforming for all the classifiers for every evaluation metric as compared to our proposed Bézier method. This indicates that the images created by the Bézier technique are of high quality in terms of classification performance as compared to the Spike2CGR-based images. Moreover, the String kernel-based results also showcase very low performance as compared to the image-based method, hence again indicating that converting sequences to images gives a more effective representation than mapping them to vectors.

## 5.2 CORONAVIRUS HOST DATASET'S PERFORMANCE

The Coronavirus host dataset-based classification performance via various evaluation metrics is reported in Appendix F.1 Table 14. We can observe that for the custom CNN models, the performance is not directly proportional to the number of hidden layers, i.e., increasing the number of hidden layers does not result in better performance, as most of the top values reside corresponding to the 1-layer CNN model and the 2-layer CNN model. This could be because the host dataset is not large enough to tackle a heavy CNN model, hence ending up having a gradient vanishing problem, which stops the model from learning. Apart from that, the ViT model is exhibiting lower performance than the custom CNN model and it can be yet again due to the dataset being small. Moreover, among the pre-trained models, ResNet-50 & VGG19 are showcasing nearly similar performance as the custom CNN classifiers (with Bézier-based images yielding maximum performance), which indicates that these models are able to generalize well using the images created by our Bézier method. However, DenseNet and EfficientNet are demonstrating very low performance for all evaluation metrics may be because the size of host data is small and these models typically need large data to attain good per-

Table 4: Classification results for different models and algorithms for **Protein Subcellular Localization dataset**. The top 5% values for each metric are underlined.

| Category | DL Model | Method | Acc. ↑ | Prec. ↑ | Recall ↑ | F1 (Weig.) ↑ | F1 (Macro) ↑ | ROC AUC ↑ | Train Time (hrs.) ↓ |
|---|---|---|---|---|---|---|---|---|---|
| Tabular Models | 3-Layer Tab CNN | OHE | 0.449 | 0.405 | 0.449 | 0.401 | 0.227 | 0.667 | 0.398 |
| | | WDGRL | 0.458 | 0.315 | 0.458 | 0.354 | 0.163 | 0.751 | 0.109 |
| | 4-Layer Tab CNN | OHE | 0.404 | 0.409 | 0.404 | 0.384 | 0.215 | 0.657 | 0.525 |
| | | WDGRL | 0.457 | 0.309 | 0.457 | 0.351 | 0.161 | 0.708 | 0.130 |
| String Kernel | - | SVM | 0.496 | 0.510 | 0.496 | 0.501 | 0.395 | 0.674 | 5.277 |
| | - | NB | 0.301 | 0.322 | 0.301 | 0.265 | 0.243 | 0.593 | 0.136 |
| | - | MLP | 0.389 | 0.390 | 0.389 | 0.388 | 0.246 | 0.591 | 7.263 |
| | - | KNN | 0.372 | 0.475 | 0.372 | 0.370 | 0.272 | 0.586 | 0.395 |
| | - | RF | 0.473 | 0.497 | 0.473 | 0.411 | 0.218 | 0.585 | 7.170 |
| | - | LR | 0.528 | 0.525 | 0.528 | 0.525 | 0.415 | 0.678 | 8.194 |
| | - | DT | 0.328 | 0.335 | 0.328 | 0.331 | 0.207 | 0.568 | 2.250 |
| Custom CNN Models | 1-Layer | FCGR | 0.545 | 0.542 | 0.545 | 0.527 | 0.386 | 0.653 | 3.065 |
| | | RandmCGR | 0.292 | 0.172 | 0.292 | 0.211 | 0.102 | 0.528 | 6.443 |
| | | Spike2CGR | 0.460 | 0.453 | 0.460 | 0.432 | 0.277 | 0.603 | 6.879 |
| | | Bézier | 0.948 | 0.919 | 0.948 | 0.931 | 0.769 | 0.890 | 3.455 |
| | % impro. of Bézier from FCGR | | 40.3 | 37.7 | 40.3 | 40.4 | 38.3 | 23.7 | -12.72 |
| | % impro. of Bézier from Spike2CGR | | 48.8 | 46.6 | 48.8 | 49.9 | 49.2 | 28.7 | 49.7 |
| | 2-Layer | FCGR | 0.565 | 0.565 | 0.565 | 0.554 | 0.432 | 0.677 | 4.074 |
| | | RandmCGR | 0.295 | 0.171 | 0.295 | 0.216 | 0.104 | 0.530 | 6.433 |
| | | Spike2CGR | 0.461 | 0.454 | 0.461 | 0.433 | 0.278 | 0.604 | 8.932 |
| | | Bézier | 0.959 | 0.971 | 0.959 | 0.963 | 0.904 | 0.965 | 13.089 |
| | % improv. of Bézier from FCGR | | 39.4 | 40.6 | 39.4 | 40.9 | 47.2 | 28.8 | -221.28 |
| | % impro. of Bézier from Spike2CGR | | 49.8 | 51.7 | 49.8 | 53 | 62.6 | 36.1 | -2922.8 |
| | 3-Layer | FCGR | 0.504 | 0.518 | 0.504 | 0.501 | 0.376 | 0.656 | 4.821 |
| | | RandmCGR | 0.303 | 0.186 | 0.303 | 0.228 | 0.110 | 0.532 | 8.930 |
| | | Spike2CGR | 0.429 | 0.430 | 0.429 | 0.421 | 0.287 | 0.612 | 3.998 |
| | | Bézier | 0.951 | 0.965 | 0.951 | 0.952 | 0.881 | 0.957 | 14.983 |
| | % improv. of Bézier from FCGR | | 44.7 | 44.7 | 44.7 | 44.8 | 50.5 | 30.1 | -210.78 |
| | % impro. of Bézier from Spike2CGR | | 52.2 | 53.5 | 52.2 | 53.1 | 59.4 | 35.5 | -274.7 |
| | 4-Layer | FCGR | 0.539 | 0.524 | 0.539 | 0.525 | 0.393 | 0.663 | 5.146 |
| | | RandmCGR | 0.311 | 0.181 | 0.311 | 0.229 | 0.110 | 0.536 | 10.234 |
| | | Spike2CGR | 0.420 | 0.420 | 0.420 | 0.424 | 0.280 | 0.600 | 9.121 |
| | | Bézier | 0.938 | 0.958 | 0.938 | 0.944 | 0.884 | 0.959 | 15.456 |
| | % improv. of Bézier from FCGR | | 39.9 | 43.4 | 39.9 | 41.9 | 49.1 | 29.6 | -200.36 |
| | % impro. of Bézier from Spike2CGR | | 51.8 | 53.8 | 51.8 | 52 | 60.4 | 35.9 | -69.4 |
| Vision Transformer | ViT | FCGR | 0.226 | 0.051 | 0.226 | 0.083 | 0.033 | 0.500 | 0.180 |
| | | RandmCGR | 0.222 | 0.049 | 0.222 | 0.080 | 0.033 | 0.500 | 0.154 |
| | | Spike2CGR | 0.222 | 0.051 | 0.222 | 0.083 | 0.147 | 0.500 | 0.176 |
| | | Bézier | 0.462 | 0.254 | 0.462 | 0.327 | 0.147 | 0.572 | 0.160 |
| | % improv. of Bézier from FCGR | | 23.6 | 20.3 | 23.6 | 24.4 | 11.4 | 7.2 | 11.11 |
| | % impro. of Bézier from Spike2CGR | | 24 | 20.3 | 24 | 24.4 | 0 | 7.2 | -9.09 |
| Pretrained Vision Models | ResNet-50 | FCGR | 0.368 | 0.268 | 0.368 | 0.310 | 0.155 | 0.556 | 3.831 |
| | | RandmCGR | 0.293 | 0.174 | 0.293 | 0.211 | 0.102 | 0.527 | 13.620 |
| | | Spike2CGR | 0.368 | 0.175 | 0.368 | 0.214 | 0.105 | 0.565 | 10.992 |
| | | Bézier | 0.964 | 0.967 | 0.964 | 0.961 | 0.907 | 0.948 | 11.415 |
| | % improv. of Bézier from FCGR | | 59.6 | 69.9 | 59.6 | 65.1 | 75.2 | 39.2 | -197.96 |
| | % impro. of Bézier from Spike2CGR | | 59.6 | 79.2 | 59.6 | 74.7 | 80.2 | 38.3 | -3.8 |
| | VGG-19 | FCGR | 0.316 | 0.209 | 0.316 | 0.241 | 0.114 | 0.533 | 14.058 |
| | | RandmCGR | 0.288 | 0.192 | 0.288 | 0.218 | 0.105 | 0.525 | 26.136 |
| | | Spike2CGR | 0.351 | 0.352 | 0.351 | 0.333 | 0.211 | 0.550 | 19.980 |
| | | Bézier | 0.896 | 0.879 | 0.896 | 0.873 | 0.680 | 0.840 | 18.837 |
| | % improv. of Bézier from FCGR | | 58 | 67 | 58 | 63.2 | 56.6 | 30.7 | -33.99 |
| | % impro. of Bézier from Spike2CGR | | 54.5 | 52.7 | 54.5 | 56.3 | 46.9 | 29 | 5.7 |
| | DenseNet | FCGR | 0.081 | 0.006 | 0.081 | 0.012 | 0.013 | 0.500 | 2.001 |
| | | RandmCGR | 0.094 | 0.008 | 0.094 | 0.016 | 0.015 | 0.500 | 1.974 |
| | | Spike2CGR | 0.099 | 0.010 | 0.099 | 0.020 | 0.002 | 0.500 | 2.111 |
| | | Bézier | 0.011 | 0.000 | 0.011 | 0.000 | 0.002 | 0.500 | 2.668 |
| | % improv. of Bézier from FCGR | | -7 | -0.6 | -7 | -1.2 | -1.1 | 0 | -33.33 |
| | % impro. of Bézier from Spike2CGR | | -8.8 | -1 | -8.8 | -2 | 0 | 0 | -26.3 |
| | EfficientNet | FCGR | 0.100 | 0.088 | 0.100 | 0.094 | 0.035 | 0.532 | 31.194 |
| | | RandmCGR | 0.284 | 0.107 | 0.284 | 0.152 | 0.078 | 0.500 | 30.223 |
| | | Spike2CGR | 0.320 | 0.230 | 0.320 | 0.230 | 0.200 | 0.500 | 25.497 |
| | | Bézier | 0.834 | 0.787 | 0.834 | 0.797 | 0.483 | 0.751 | 20.312 |
| | % improv. of Bézier from FCGR | | 73.4 | 69.9 | 73.4 | 70.3 | 44.8 | 21.9 | 34.88 |
| | % impro. of Bézier from Spike2CGR | | 51.4 | 55.7 | 51.4 | 56.7 | 28.3 | 25.1 | 20.3 |

formance. Additionally, the feature-engineering-based methods lean towards a lower performance bound for all the evaluation metrics corresponding to both 3-layer Tab CNN & 4-layer Tab CNN, and most of the ML classifiers based on the String kernel also showcase less performance. This indicates that converting the host sequences into images can preserve more relevant information in the respective images about the sequence in terms of classification performance as compared to converting them into vectors. Furthermore, among the image generation methods, RandomCGR has the lowest performance for every metric while Bézier (our method), Spike2CGR, and FCGR have comparable performance as they yield most of the top values for all the metrics. Overall, Bézier seems to perform well for the host classification task, implying that the images generated by it are of good quality for classification.

## 5.3 ACP DATASET'S PERFORMANCE

The classification performance achieved using the ACP dataset for various evaluation metrics is summarized in Appendix F.2 Table 15. We can observe that increasing the number of inner layers for the custom CNN models does not enhance the predictive performance, as 1-layer CNN & 2-layer CNN models portray higher performance. This could be because the ACP dataset is very small, so using a large model can cause a gradient vanishing challenge and, hence, hinder the learning process. Additionally, the ViT model is yielding lower performance than the custom CNN models and it can be due to yet again the dataset being very small. Moreover, the pre-trained ResNet-50 and VGG19 models depict very similar performance as the custom CNN models. This shows that the ResNet and VGG19 models are able to generalize well to our Bézier-based data. However, the EfficeintNet and Denset classifiers portray very low performance for every evaluation metric. It can be due to their architectures which require large data for fine-tuning the model, however, our dataset is extremely small. Furthermore, the feature-engineering-based embedding approaches are overall showcasing bad performance (except for 4 tab CNN OHE) as compared to the image-based methods. It implies that the bio-sequences's information is effectively preserved in the respective image form rather than the vector form generated from the feature-engineering methods in terms of predictive performance. Note that, although the String kernel embedding-based ML classifiers are yielding the highest performances corresponding to every evaluation metric, our method's performance is also close to it, which means that our method is also yielding an effective representation for sequences. For the image-based embedding methods, we can notice that our method (Bézier) and the FCGR baselines illustrate comparable predictive results, while RandomCGR and Spike2CGR lean toward the lower performance bound. Overall, we can claim that the Bézier method exhibits good performance for the ACP classification task.

## 5.4 HUMAN DNA DATASET PERFORMANCE

The classification results for the DL model using the Human DNA dataset are given in Table 5. We can observe that the pre-trained vision models and the vision transformer classifier are yielding very low performance corresponding to every image-based strategy. It can be again due to the gradient vanishing problem because of the small size of the dataset. Moreover, the customer CNN models are obtaining high performance, especially for the 1-layer CNN model and 2-layer CNN model. Note that increasing the number of layers in the custom CNN models is reducing the performance, and a small dataset could be a reason for this behavior too. We can also notice that our proposed Bézier method is able to achieve performance in the top $5\%$ for almost every evaluation metric corresponding to the custom CNN classifiers. Furthermore, the image-based methods clearly outperform the feature-engineering ones, hence indicating that converting the nucleotide sequences to images can retain more information about the sequences as compared to mapping them to vectors in terms of classification predictive performance. Similarly, the String kernel method-based ML classifiers, except RF, also portray less performance than the custom CNN models which yet again proves that converting sequences into images is more effective than mapping them to vectors.

## 5.5 SMILES STRING DATASET PERFORMANCE

Table 5: Classification results for different models and algorithms for **Human DNA dataset**. The top 5% values for each metric are underlined.

| Category | DL Model | Method | Acc. ↑ | Prec. ↑ | Recall ↑ | F1 (Weig.) ↑ | F1 (Macro) ↑ | ROC AUC ↑ | Train Time (hrs.) ↓ |
|---|---|---|---|---|---|---|---|---|---|
| Tabular Models | 3-Layer Tab CNN | OHE | 0.627 | 0.699 | 0.627 | 0.613 | 0.566 | 0.729 | 0.024 |
| | | WDGRL | 0.657 | 0.716 | 0.657 | 0.649 | 0.601 | 0.758 | 0.020 |
| | 4-Layer Tab CNN | OHE | 0.680 | 0.704 | 0.680 | 0.661 | 0.581 | 0.762 | 0.042 |
| | | WDGRL | 0.654 | 0.692 | 0.654 | 0.635 | 0.551 | 0.743 | 0.038 |
| String Kernel | - | SVM | 0.618 | 0.617 | 0.618 | 0.613 | 0.588 | 0.753 | 39.791 |
| | - | NB | 0.338 | 0.452 | 0.338 | 0.347 | 0.333 | 0.617 | 0.276 |
| | - | MLP | 0.597 | 0.595 | 0.597 | 0.593 | 0.549 | 0.737 | 331.068 |
| | - | KNN | 0.645 | 0.657 | 0.645 | 0.646 | 0.612 | 0.774 | 1.274 |
| | - | RF | 0.731 | 0.776 | 0.731 | 0.729 | 0.723 | 0.808 | 12.673 |
| | - | LR | 0.571 | 0.570 | 0.571 | 0.558 | 0.532 | 0.716 | 2.995 |
| | - | DT | 0.630 | 0.631 | 0.630 | 0.630 | 0.598 | 0.767 | 2.682 |
| Custom CNN Models | 1-Layer | FCGR | 0.717 | 0.719 | 0.717 | 0.709 | 0.711 | 0.834 | 0.351 |
| | | RandmCGR | 0.820 | 0.827 | 0.820 | 0.816 | 0.787 | 0.872 | 0.355 |
| | | Spike2CGR | 0.662 | 0.698 | 0.662 | 0.660 | 0.627 | 0.768 | 0.353 |
| | | Bézier | 0.710 | 0.712 | 0.710 | 0.700 | 0.713 | 0.831 | 0.339 |
| | % impro. of Bézier from Spike2CGR | | 5.1 | 1.4 | 5.1 | 4 | 8.6 | 6.3 | -3.9 |
| | 2-Layer | FCGR | 0.705 | 0.708 | 0.705 | 0.694 | 0.691 | 0.831 | 0.365 |
| | | RandmCGR | 0.785 | 0.791 | 0.785 | 0.782 | 0.750 | 0.845 | 0.622 |
| | | Spike2CGR | 0.665 | 0.685 | 0.665 | 0.664 | 0.633 | 0.786 | 0.692 |
| | | Bézier | 0.700 | 0.722 | 0.700 | 0.695 | 0.659 | 0.803 | 0.350 |
| | % impro. of Bézier from Spike2CGR | | 3.5 | 3.7 | 3.5 | 3.1 | 2.6 | 1.7 | 49.4 |
| | 3-Layer | FCGR | 0.632 | 0.641 | 0.632 | 0.623 | 0.609 | 0.767 | 0.332 |
| | | RandmCGR | 0.710 | 0.724 | 0.710 | 0.697 | 0.661 | 0.807 | 0.530 |
| | | Spike2CGR | 0.580 | 0.636 | 0.580 | 0.582 | 0.514 | 0.715 | 0.331 |
| | | Bézier | 0.426 | 0.498 | 0.426 | 0.351 | 0.298 | 0.594 | 0.376 |
| | % impro. of Bézier from Spike2CGR | | -15.4 | -13.8 | -15.4 | -23.1 | -21.6 | -12.1 | -13.59 |
| | 4-Layer | FCGR | 0.300 | 0.090 | 0.300 | 0.138 | 0.065 | 0.500 | 0.331 |
| | | RandmCGR | 0.287 | 0.082 | 0.287 | 0.128 | 0.063 | 0.500 | 0.521 |
| | | Spike2CGR | 0.377 | 0.385 | 0.377 | 0.305 | 0.232 | 0.562 | 0.311 |
| | | Bézier | 0.313 | 0.097 | 0.313 | 0.149 | 0.068 | 0.500 | 0.321 |
| | % impro. of Bézier from Spike2CGR | | -6.4 | -28.8 | -6.4 | -15.6 | -16.4 | -6.2 | -3.2 |
| Vision Transformer | ViT | FCGR | 0.300 | 0.090 | 0.300 | 0.138 | 0.065 | 0.500 | 0.782 |
| | | RandmCGR | 0.295 | 0.140 | 0.295 | 0.142 | 0.097 | 0.510 | 0.828 |
| | | Spike2CGR | 0.307 | 0.094 | 0.307 | 0.144 | 0.067 | 0.500 | 3.787 |
| | | Bézier | 0.382 | 0.326 | 0.382 | 0.323 | 0.239 | 0.613 | 0.654 |
| | % impro. of Bézier from Spike2CGR | | 7.5 | 23.2 | 7.5 | 17.9 | 17.2 | 11.3 | 82.7 |
| Pretrained Vision Models | ResNet-50 | FCGR | 0.357 | 0.251 | 0.357 | 0.283 | 0.208 | 0.500 | 0.495 |
| | | RandmCGR | 0.290 | 0.192 | 0.290 | 0.137 | 0.072 | 0.500 | 0.481 |
| | | Spike2CGR | 0.352 | 0.341 | 0.352 | 0.295 | 0.208 | 0.565 | 2.443 |
| | | Bézier | 0.408 | 0.244 | 0.408 | 0.294 | 0.184 | 0.561 | 0.873 |
| | % impro. of Bézier from Spike2CGR | | 5.6 | -9.7 | 5.6 | -0.1 | -2.4 | -0.4 | 64.2 |
| | VGG-19 | FCGR | 0.345 | 0.285 | 0.345 | 0.249 | 0.181 | 0.540 | 1.078 |
| | | RandmCGR | 0.287 | 0.082 | 0.287 | 0.128 | 0.063 | 0.500 | 1.115 |
| | | Spike2CGR | 0.307 | 0.094 | 0.307 | 0.144 | 0.067 | 0.500 | 3.032 |
| | | Bézier | 0.317 | 0.132 | 0.317 | 0.176 | 0.098 | 0.510 | 1.221 |
| | % impro. of Bézier from Spike2CGR | | 1 | 3.8 | 1 | 3.2 | 3.1 | 1 | 59.7 |
| | DenseNet | FCGR | 0.075 | 0.005 | 0.075 | 0.010 | 0.019 | 0.500 | 0.764 |
| | | RandmCGR | 0.062 | 0.003 | 0.062 | 0.007 | 0.016 | 0.500 | 0.825 |
| | | Spike2CGR | 0.067 | 0.004 | 0.067 | 0.008 | 0.018 | 0.500 | 1.295 |
| | | Bézier | 0.078 | 0.007 | 0.078 | 0.013 | 0.022 | 0.491 | 0.822 |
| | % impro. of Bézier from Spike2CGR | | 1.1 | 0.3 | 1.1 | 0.5 | 0.4 | -0.9 | 36.5 |
| | EfficientNet | FCGR | 0.200 | 0.141 | 0.200 | 0.147 | 0.094 | 0.517 | 0.814 |
| | | RandmCGR | 0.287 | 0.082 | 0.287 | 0.128 | 0.063 | 0.500 | 0.837 |
| | | Spike2CGR | 0.275 | 0.169 | 0.275 | 0.187 | 0.112 | 0.524 | 1.343 |
| | | Bézier | 0.313 | 0.097 | 0.313 | 0.149 | 0.068 | 0.500 | 0.844 |
| | % impro. of Bézier from Spike2CGR | | 3.8 | -7.2 | 3.8 | -3.8 | -4.4 | -2.4 | 37.1 |

The classification results for the DL model using the SMILES String dataset are given in Table 6. We can observe that, the performance achieved by all the classifiers corresponding to every embedding strategy (image or vector) is very good and similar to each other, except for the DenseNet and EfficientNet models which have bad results. A reason for the bad results could be the small size of the

data as DenseNet and EfficientNet usually operate on large datasets to have optimal performance. Note that, although most of the classifiers portray similar results, our method achieves the maximum performance. Moreover, as this data contains sequences constituted of more than 20 unique characters, therefore, the FCGR & Spike2CGR methods failed to operate on them. Furthermore, our image-based method is performing better than the tabular ones (feature-engineering-based and String kernel-based), hence obtaining images of sequences is more useful for the classification tasks.

Table 6: Classification results for different models and algorithms for **SMILES String dataset**. The best value for each metric is underlined. As the performances of most of the models are the same and highlighting the top 5% includes a lot of data, that's why we only underlined the best one.

| Category | DL Model | Method | Acc. ↑ | Prec. ↑ | Recall ↑ | F1 (Weig.) ↑ | F1 (Macro) ↑ | ROC AUC ↑ | Train Time (hrs.) ↓ |
|---|---|---|---|---|---|---|---|---|---|
| Tabular Models | 3-Layer Tab CNN | OHE | 0.966 | 0.935 | 0.966 | 0.950 | 0.098 | 0.500 | 0.132 |
| | | WDGRL | 0.966 | 0.935 | 0.966 | 0.950 | 0.098 | 0.500 | 0.001 |
| | 4-Layer Tab CNN | OHE | 0.966 | 0.935 | 0.966 | 0.950 | 0.098 | 0.500 | 0.155 |
| | | WDGRL | 0.966 | 0.935 | 0.966 | 0.950 | 0.098 | 0.500 | 0.001 |
| String Kernel | - | SVM | 0.812 | 0.813 | 0.812 | 0.811 | 0.084 | 0.502 | 10.254 |
| | - | NB | 0.537 | 0.643 | 0.537 | 0.549 | 0.096 | 0.502 | 1.24 |
| | - | MLP | 0.789 | 0.788 | 0.789 | 0.790 | 0.079 | 0.505 | 13.149 |
| | - | KNN | 0.844 | 0.858 | 0.844 | 0.842 | 0.087 | 0.503 | 2.348 |
| | - | RF | 0.929 | 0.927 | 0.929 | 0.925 | 0.098 | 0.507 | 9.315 |
| | - | LR | 0.772 | 0.769 | 0.772 | 0.760 | 0.073 | 0.502 | 5.652 |
| | - | DT | 0.834 | 0.829 | 0.834 | 0.832 | 0.075 | 0.508 | 3.318 |
| Custom CNN Models | 1-Layer | RandmCGR | 0.962 | 0.926 | 0.962 | 0.944 | 0.098 | 0.500 | 0.988 |
| | | Bézier | 0.970 | 0.942 | 0.970 | 0.956 | 0.109 | 0.512 | 1.003 |
| | % impro. of Bézier from RandomCGR | | 0.8 | 1.6 | 0.8 | 1.2 | 1.1 | 12 | -1.51 |
| | 2-Layer | RandmCGR | 0.962 | 0.926 | 0.962 | 0.944 | 0.098 | 0.500 | 0.989 |
| | | Bézier | 0.970 | 0.942 | 0.970 | 0.956 | 0.109 | 0.512 | 1.253 |
| | % impro. of Bézier from RandomCGR | | 0.8 | 1.6 | 0.8 | 1.2 | 1.1 | 12 | -26.6 |
| | 3-Layer | RandmCGR | 0.962 | 0.926 | 0.962 | 0.944 | 0.098 | 0.500 | 1.411 |
| | | Bézier | 0.970 | 0.942 | 0.970 | 0.956 | 0.109 | 0.511 | 1.082 |
| | % impro. of Bézier from RandomCGR | | 0.8 | 1.6 | 0.8 | 1.2 | 1.1 | 12 | 80.04 |
| | 4-Layer | RandmCGR | 0.962 | 0.926 | 0.962 | 0.944 | 0.098 | 0.500 | 1.331 |
| | | Bézier | 0.970 | 0.942 | 0.970 | 0.956 | 0.109 | 0.512 | 1.210 |
| | % impro. of Bézier from RandomCGR | | 0.8 | 1.6 | 0.8 | 1.2 | 1.1 | 12 | 9.09 |
| Vision Transformer | ViT | RandmCGR | 0.962 | 0.926 | 0.962 | 0.944 | 0.098 | 0.500 | 1.876 |
| | | Bézier | 0.970 | 0.942 | 0.970 | 0.956 | 0.109 | 0.512 | 1.864 |
| | % impro. of Bézier from RandomCGR | | 0.8 | 1.6 | 0.8 | 1.2 | 1.1 | 12 | 0.63 |
| Pretrained Vision Models | ResNet-50 | RandmCGR | 0.962 | 0.926 | 0.962 | 0.944 | 0.098 | 0.500 | 1.872 |
| | | Bézier | 0.970 | 0.940 | 0.970 | 0.950 | 0.100 | 0.500 | 1.142 |
| | % impro. of Bézier from RandomCGR | | 0.8 | 1.4 | 0.8 | 0.6 | 0.2 | 0 | 38.99 |
| | VGG-19 | RandmCGR | 0.962 | 0.926 | 0.962 | 0.944 | 0.098 | 0.500 | 7.120 |
| | | Bézier | 0.970 | 0.940 | 0.970 | 0.950 | 0.100 | 0.500 | 2.899 |
| | % impro. of Bézier from RandomCGR | | 0.8 | 1.4 | 0.8 | 0.6 | 0.2 | 0 | 59.2 |
| | DenseNet | RandmCGR | 0.001 | 0.024 | 0.001 | 0.004 | 0.000 | 0.500 | 5.043 |
| | | Bézier | 0.001 | 0.023 | 0.001 | 0.066 | 0.000 | 0.500 | 2.867 |
| | % impro. of Bézier from RandomCGR | | 0 | 1 | 0 | 6.2 | 0 | 0 | 43.14 |
| | EfficientNet | RandmCGR | 0.962 | 0.926 | 0.962 | 0.944 | 0.098 | 0.500 | 4.892 |
| | | Bézier | 0.969 | 0.938 | 0.969 | 0.950 | 0.100 | 0.500 | 3.892 |
| | % impro. of Bézier from RandomCGR | | 0.6 | 1.2 | 0.6 | 5.6 | 0.2 | 0 | 20.44 |

## 5.6 MUSIC GENRE DATASET PERFORMANCE

The classification results for the DL model using the Music Genre dataset are given in Table 7. An important point to note here is that since the number of unique characters in the music data is > 20, the traditional FCGR and Spike2CGR methods fail to run on such datasets. In general, although the RandomCGR method performs better using classical vision models, the performance drastically reduces compared to the proposed method on the pre-trained vision models (e.g. see results for VGG-19 results in Table 7). Such behavior supports our argument that in general, the proposed method improves the performance of the pre-trained models in terms of sequence classification. Moreover, the image-based methods are clearly outperforming the feature-engineering and String-kernel baselines, hence image representations are more promising for doing classification than the tabular ones.

Table 7: Classification results for different models and algorithms for **Music Genre dataset**. The top 5% values for each metric are underlined.

| Category | DL Model | Method | Acc. ↑ | Prec. ↑ | Recall ↑ | F1 (Weig.) ↑ | F1 (Macro) ↑ | ROC AUC ↑ | Train Time (hrs.) ↓ |
|---|---|---|---|---|---|---|---|---|---|
| Tabular Models | 3-Layer Tab CNN | OHE | 0.515 | 0.682 | 0.515 | 0.611 | 0.451 | 0.652 | 0.003 |
| | | WDGRL | 0.599 | 0.600 | 0.599 | 0.565 | 0.515 | 0.659 | 0.001 |
| | 4-Layer Tab CNN | OHE | 0.516 | 0.555 | 0.516 | 0.511 | 0.451 | 0.652 | 0.003 |
| | | WDGRL | 0.612 | 0.588 | 0.612 | 0.588 | 0.530 | 0.670 | 0.001 |
| String Kernel | - | SVM | 0.301 | 0.322 | 0.301 | 0.294 | 0.294 | 0.615 | 0.886 |
| | - | NB | 0.369 | 0.376 | 0.369 | 0.357 | 0.352 | 0.649 | 0.039 |
| | - | MLP | 0.219 | 0.231 | 0.219 | 0.212 | 0.211 | 0.568 | 3.476 |
| | - | KNN | 0.400 | 0.409 | 0.400 | 0.388 | 0.387 | 0.669 | 0.169 |
| | - | RF | 0.341 | 0.354 | 0.341 | 0.334 | 0.333 | 0.638 | 1.478 |
| | - | LR | 0.397 | 0.397 | 0.397 | 0.389 | 0.386 | 0.666 | 21.209 |
| | - | DT | 0.283 | 0.290 | 0.283 | 0.282 | 0.281 | 0.603 | 0.392 |
| Custom CNN Models | 1-Layer | RandmCGR | 0.989 | 0.989 | 0.989 | 0.989 | 0.989 | 0.989 | 0.400 |
| | | Bézier | 0.957 | 0.953 | 0.957 | 0.953 | 0.844 | 0.919 | 0.312 |
| | % improv. of Bézier from RandomCGR | | -3.2 | -3.6 | -3.2 | -3.6 | -14.5 | -7 | 22 |
| | 2-Layer | RandmCGR | 0.985 | 0.985 | 0.985 | 0.985 | 0.985 | 0.992 | 0.4121 |
| | | Bézier | 0.943 | 0.941 | 0.943 | 0.939 | 0.827 | 0.911 | 0.345 |
| | % improv. of Bézier from RandomCGR | | -4.2 | -4.3 | -4.2 | -4.6 | -15.8 | -1.1 | 16.2 |
| | 3-Layer | RandmCGR | 0.085 | 0.007 | 0.085 | 0.013 | 0.015 | 0.500 | 0.541 |
| | | Bézier | 0.886 | 0.893 | 0.886 | 0.882 | 0.789 | 0.887 | 0.453 |
| | % improv. of Bézier from RandomCGR | | 80.1 | 88.6 | 80.1 | 86.9 | 77.4 | 38.7 | 16.2 |
| | 4-Layer | RandmCGR | 0.155 | 0.044 | 0.155 | 0.063 | 0.074 | 0.545 | 0.554 |
| | | Bézier | 0.900 | 0.908 | 0.900 | 0.897 | 0.802 | 0.895 | 0.438 |
| | % improv. of Bézier from RandomCGR | | 74.5 | 86.4 | 74.5 | 83.4 | 72.8 | 35 | 20.9 |
| Vision Transformer | ViT | RandmCGR | 0.110 | 0.012 | 0.110 | 0.021 | 0.019 | 0.500 | 0.807 |
| | | Bézier | 0.099 | 0.009 | 0.099 | 0.017 | 0.022 | 0.500 | 1.090 |
| | % improv. of Bézier from RandomCGR | | -1.1 | -0.3 | -1.1 | -0.4 | -0.3 | 0 | -23.9 |
| Pretrained Vision Models | ResNet-50 | RandmCGR | 0.525 | 0.608 | 0.525 | 0.485 | 0.496 | 0.740 | 0.653 |
| | | Bézier | 0.546 | 0.545 | 0.546 | 0.479 | 0.457 | 0.728 | 0.543 |
| | % improv. of Bézier from RandomCGR | | 2.1 | -6.3 | 2.1 | 0.6 | -3.9 | -1.2 | 16.8 |
| | VGG-19 | RandmCGR | 0.410 | 0.421 | 0.410 | 0.334 | 0.410 | 0.673 | 1.220 |
| | | Bézier | 0.843 | 0.867 | 0.843 | 0.838 | 0.741 | 0.856 | 1.421 |
| | % improv. of Bézier from RandomCGR | | 43.3 | 44.6 | 43.3 | 50.4 | 33.1 | 18.3 | -16.47 |
| | DenseNet | RandmCGR | 0.080 | 0.056 | 0.080 | 0.052 | 0.053 | 0.489 | 2.118 |
| | | Bézier | 0.113 | 0.130 | 0.113 | 0.043 | 0.049 | 0.508 | 2.332 |
| | % improv. of Bézier from RandomCGR | | 3.3 | 7.4 | 3.3 | -0.9 | -0.4 | 1.9 | -10.10 |
| | EfficientNet | RandmCGR | 0.735 | 0.719 | 0.735 | 0.697 | 0.689 | 0.851 | 1.011 |
| | | Bézier | 0.929 | 0.928 | 0.929 | 0.924 | 0.808 | 0.898 | 0.889 |
| | % improv. of Bézier from RandomCGR | | 19.4 | 20.9 | 19.4 | 22.7 | 11.9 | 4.7 | 12.06 |

## 5.7 T-SNE DATA VISUALIZATION

We visualized the feature vectors using the t-SNE (t-distributed stochastic neighbor embedding) (Van der Maaten & Hinton, 2008) approach extracted from the last hidden layer of the 2-layer

CNN model for each of our datasets. The plots are computed for the images generated by the FCGR baseline and our proposed Bézier method.

The t-SNE visualization of FCGR and Bézier images of the protein subcellular localization dataset is illustrated in Figure 2. We can clearly observe that the clusters generated corresponding to the Bézier data are very defined and visible. It indicates that the data structure is highly preserved even in 2D space due to the high quality of the respective embeddings used. As these embeddings are acquired from the images generated by our Bézier method, it implies that the images constructed by our method are of high quality and contain the sequence information efficiently and effectively. However, the t-SNE plot against the FCGR method consists of very overlapping and non-definite clusters, which indicates that the FCGR-based embeddings are unable to retain a good cluster structure in a low dimensional space, hence they are suboptimal. Moreover, the t-SNE plots of the Coronavirus host dataset and ACP dataset are given in Appendix F.3 along with their respective discussions.

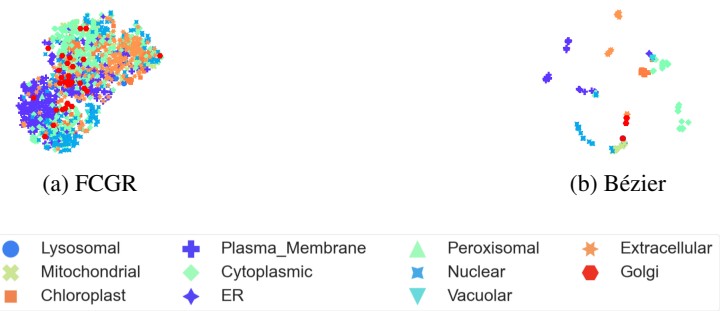

(a) FCGR                    (b) Bézier

| ● Lysosomal | ✚ Plasma_Membrane | ▲ Peroxisomal | ✳ Extracellular |
| ✖ Mitochondrial | ◆ Cytoplasmic | ◄ Nuclear | ⬣ Golgi |
| ■ Chloroplast | ◆ ER | ▼ Vacuolar | |

Figure 2: The t-SNE plots of **Protein Subcellular Localization dataset** embeddings extracted from the last layer of **2layer CNN** classifier using the FCGR- and Bézier-based images respectively. The figure is best seen in color.

## 5.8 CONFUSION MATRIX RESULTS AND DISCUSSION

We investigated the confusion matrices obtained from the respective test sets of our host and protein subcellular datasets corresponding to the 2-layer CNN model for the FCGR baseline method and our proposed Bézier technique. We chose the 2-layer CNN classifier because it contains mostly the optimal predictive performance values for every dataset.

The confusion matrices corresponding to the protein subcellular localization dataset are illustrated in Figure 3. We can observe that our method is tremendously outperforming the FCGR baseline strategy as it has optimal true positive counts. Moreover, Bézier is also able to attain high performance for each category of the dataset. Overall, we can witness that our method has almost perfect performance for the protein subcellular localization classification task. Furthermore, the confusion matrices for the host dataset are given in Appendix F.6 Figure 20.

## 6 CONCLUSION

In this work, we proposed a novel technique to convert biological sequences into images using the Bézier curve. It enables us to apply the sophisticated DL vision classifiers in the analysis of biological sequences. We validated our idea using three distinct protein datasets, and our method tremendously outperforms the baselines for protein subcellular localization classification and shows good performance on other dataset classifications. In the future, we want to explore the scalability of our technique by applying it to larger datasets. Moreover, we also want to investigate the generalizability of our method by using it on nucleotide-based datasets in future.

## REFERENCES

Sarwan Ali and Murray Patterson. Spike2vec: An efficient and scalable embedding approach for covid-19 spike sequences. In *International Conference on Big Data (Big Data)*, pp. 1533–1540,

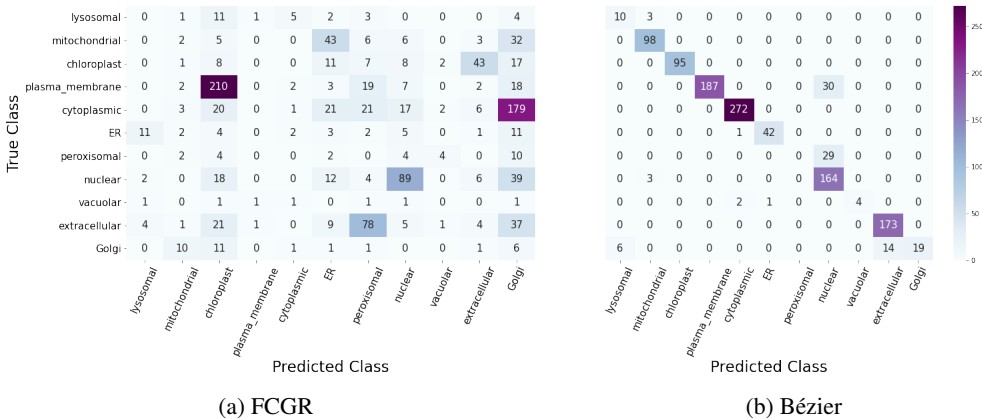

Figure 3: Confusion matrices of **Protein Subcellular Localization dataset** for 2layer CNN classifier using the FCGR and Bézier image generation methods.

2021.

Sarwan Ali, Bikram Sahoo, Muhammad Asad Khan, Alexander Zelikovsky, Imdad Ullah Khan, and Murray Patterson. Efficient approximate kernel based spike sequence classification. *IEEE/ACM Transactions on Computational Biology and Bioinformatics*, 2022.

Senay Baydas and Bulent Karakas. Defining a curve as a bezier curve. *Journal of Taibah University for Science*, 13(1):522–528, 2019.

Nadav Brandes, Dan Ofer, Yam Peleg, Nadav Rappoport, and Michal Linial. ProteinBERT: a universal deep-learning model of protein sequence and function. *Bioinformatics*, 38(8):2102–2110, 02 2022. ISSN 1367-4803. doi: 10.1093/bioinformatics/btac020. URL https://doi.org/10.1093/bioinformatics/btac020.

M. Farhan, J. Tariq, A. Zaman, M. Shabbir, and I. Khan. Efficient approximation algorithms for strings kernel based sequence classification. In *Advances in neural information processing systems (NeurIPS)*, pp. 6935–6945. ., 2017.

GISAID Website. https://www.gisaid.org/, 2021. [Online; accessed 29-December-2021].

Grisoni et al. 'de novo design of anticancer peptides by ensemble artificial neural networks'. *'Journal of Molecular Modeling'*, '25'('5'):'112', 'Apr' '2019'. ISSN '0948-5023'. doi: '10.1007/s00894-019-4007-6'. URL '[WebLink]'.

J. Hadfield, C. Megill, S.M. Bell, J. Huddleston, B. Potter, C. Callender, P. Sagulenko, T. Bedford, and R.A. Neher. Nextstrain: real-time tracking of pathogen evolution. *Bioinformatics*, 34:4121–4123, 2018.

Xi-An Han, YiChen Ma, and XiLi Huang. A novel generalization of bézier curve and surface. *Journal of Computational and Applied Mathematics*, 217(1):180–193, 2008.

Kaiming He, Xiangyu Zhang, Shaoqing Ren, and Jian Sun. Deep residual learning for image recognition. In *IEEE conference on computer vision and pattern recognition*, pp. 770–778, 2016.

Michael Heinzinger, Ahmed Elnaggar, Yu Wang, Christian Dallago, Dmitrii Nechaev, Florian Matthes, and Burkhard Rost. Modeling aspects of the language of life through transfer-learning protein sequences. *BMC bioinformatics*, 20(1):1–17, 2019.

Takatsug Hirokawa, Seah Boon-Chieng, and Shigeki Mitaku. Sosui: classification and secondary structure prediction system for membrane proteins. *Bioinformatics (Oxford, England)*, 14(4): 378–379, 1998.

Ronny Hug, Wolfgang Hübner, and Michael Arens. Introducing probabilistic bézier curves for n-step sequence prediction. In *Proceedings of the AAAI Conference on Artificial Intelligence*, volume 34, pp. 10162–10169, 2020.

Human DNA. https://www.kaggle.com/code/nageshsingh/demystify-dna-sequencing-with-machine-learning/data. [Online; accessed 10-October-2022].

Forrest Iandola, Matt Moskewicz, Sergey Karayev, Ross Girshick, Trevor Darrell, and Kurt Keutzer. Densenet: Implementing efficient convnet descriptor pyramids. *arXiv preprint arXiv:1404.1869*, 2014.

H Joel Jeffrey. Chaos game representation of gene structure. *Nucleic acids research*, 18(8):2163–2170, 1990.

Kiril Kuzmin et al. Machine learning methods accurately predict host specificity of coronaviruses based on spike sequences alone. *Biochemical and Biophysical Research Communications*, 533 (3):553–558, 2020.

Tao Li, Mitsunori Ogihara, and Qi Li. A comparative study on content-based music genre classification. In *Proceedings of the 26th annual international ACM SIGIR conference on Research and development in informaion retrieval*, pp. 282–289, 2003.

Yuliang Liu, Chunhua Shen, Lianwen Jin, Tong He, Peng Chen, Chongyu Liu, and Hao Chen. Abcnet v2: Adaptive bezier-curve network for real-time end-to-end text spotting. *IEEE Transactions on Pattern Analysis and Machine Intelligence*, 44(11):8048–8064, 2021.

Hannah F Löchel, Dominic Eger, Theodor Sperlea, and Dominik Heider. Deep learning on chaos game representation for proteins. *Bioinformatics*, 36(1):272–279, 2020.

Yuanlin Ma, Zuguo Yu, Runbin Tang, Xianhua Xie, Guosheng Han, and Vo V Anh. Phylogenetic analysis of hiv-1 genomes based on the position-weighted k-mers method. *Entropy*, 22(2):255, 2020.

B. Q. Minh et al. Iq-tree 2: New models and efficient methods for phylogenetic inference in the genomic era. *Molecular Biology and Evolution*, 37(5):1530–1534, 2020.

Taslim Murad, Sarwan Ali, and Murray Patterson. A new direction in membranolytic anticancer peptides classification: Combining spaced k-mers with chaos game representation.

Taslim Murad, Sarwan Ali, Imdadullah Khan, and Murray Patterson. Spike2cgr: an efficient method for spike sequence classification using chaos game representation. *Machine Learning*, pp. 1–26, 2023.

Brett E Pickett, Eva L Sadat, Yun Zhang, Jyothi M Noronha, R Burke Squires, Victoria Hunt, Mengya Liu, Sanjeev Kumar, Sam Zaremba, Zhiping Gu, et al. Vipr: an open bioinformatics database and analysis resource for virology research. *Nucleic acids research*, 40(D1):D593–D598, 2012.

Protein Subcellular Localization. https://www.kaggle.com/datasets/lzyacht/proteinsubcellularlocalization, 2022. [Online; accessed 10-October-2022].

Limeng Qiao, Wenjie Ding, Xi Qiu, and Chi Zhang. End-to-end vectorized hd-map construction with piecewise bezier curve. In *Proceedings of the IEEE/CVF Conference on Computer Vision and Pattern Recognition*, pp. 13218–13228, 2023.

Didier Rognan. Chemogenomic approaches to rational drug design. *British journal of pharmacology*, 152(1):38–52, 2007.

Yosi Shamay, Janki Shah, Mehtap Işık, Aviram Mizrachi, Josef Leibold, Darjus F Tschaharganeh, Daniel Roxbury, Januka Budhathoki-Uprety, Karla Nawaly, James L Sugarman, et al. Quantitative self-assembly prediction yields targeted nanomedicines. *Nature materials*, 17(4):361–368, 2018.

Jian Shen, Yanru Qu, Weinan Zhang, and Yong Yu. Wasserstein distance guided representation learning for domain adaptation. In *AAAI conference on artificial intelligence*, 2018.

Karen Simonyan and Andrew Zisserman. Very deep convolutional networks for large-scale image recognition. In *International Conference on Learning Representations*, 2015.

Nils Strodthoff, Patrick Wagner, Markus Wenzel, and Wojciech Samek. Udsmprot: universal deep sequence models for protein classification. *Bioinformatics*, 36(8):2401–2409, 2020.

Mingxing Tan and Quoc Le. Efficientnet: Rethinking model scaling for convolutional neural networks. In *International conference on machine learning*, pp. 6105–6114. PMLR, 2019.

Laurens Van der Maaten and Geoffrey Hinton. Visualizing data using t-sne. *Journal of machine learning research*, 9(11), 2008.

James C Whisstock and Arthur M Lesk. Prediction of protein function from protein sequence and structure. *Quarterly reviews of biophysics*, 36(3):307–340, 2003.

Junyuan Xie, Ross Girshick, and Ali Farhadi. Unsupervised deep embedding for clustering analysis. In *International conference on machine learning*, pp. 478–487, 2016.

Yao et al. Negative log likelihood ratio loss for deep neural network classification. In *Proceedings of the Future Technologies Conference*, pp. 276–282. Springer, 2019.

Michaela Areti Zervou, Effrosyni Doutsi, Pavlos Pavlidis, and Panagiotis Tsakalides. Structural classification of proteins based on the computationally efficient recurrence quantification analysis and horizontal visibility graphs. *Bioinformatics*, 37(13):1796–1804, 2021.

## A   INTRODUCTION

Figure 4 demonstrates the recursive procedures followed by CGR (Jeffrey, 1990), FCGR (Löchel et al., 2020), and protein secondary structure prediction (Zervou et al., 2021) to allocate a location to nucleotides/amino acids in the corresponding generated images respectively.

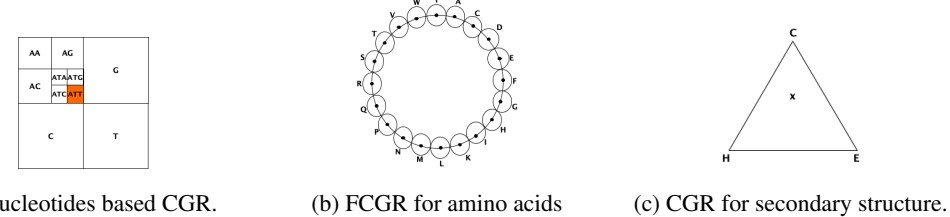

(a) Nucleotides based CGR.  (b) FCGR for amino acids  (c) CGR for secondary structure.

Figure 4: Figure (a) shows the CGR (Jeffrey, 1990) based determination of location for the "ATT" nucleotide sequence in the respective image. Figure (b) illustrates the 20-flakes-based image created using the FCGR (Löchel et al., 2020) method for a sequence of amino acids. (c) shows the CGR representation for the secondary protein structure (Zervou et al., 2021).

## B   PROPOSED APPROACH

Two example images generated by the Bézier curve-based method for two sequences (one from the active class and the other from the inactive class) from the ACP dataset are illustrated in Figure 5. We can observe that the created images are different from each other, which indicates that our method is able to preserve the information possessed by the sequences differently in the respective generated images.

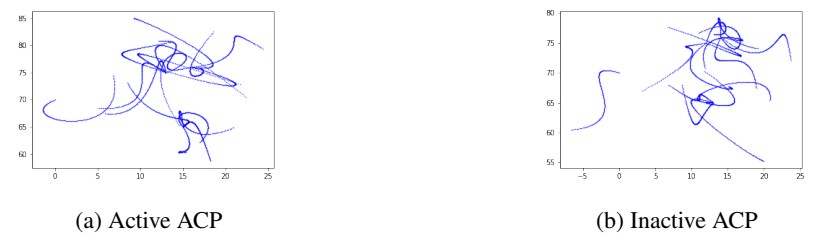

(a) Active ACP  (b) Inactive ACP

Figure 5: The Bézier curve method-based images created for two sequences from the ACP dataset. One sequence belongs to the active class of the dataset, while the other is from the inactive class.

## C   DATA STATISTICS

We have used 3 distinct protein sequence datasets to evaluate our proposed system. The details of each one are given as follows,

### C.1   PROTEIN SUBCELLULAR LOCALIZATION DATASET

This dataset (Protein Subcellular Localization, 2022) has 5959 unaligned protein sequences distributed among 11 unique subcellular locations. The associated subcellular location is predicted for a given protein sequence as input. The statistical details of the data are illustrated in Table 8.

Table 8: The protein subcellular localization dataset distribution with respect to the subcellular locations. The minimum, maximum, and average lengths of sequences belonging to each subcellular location are also mentioned.

| Subcellular Locations | Count | Protein Subcellular Sequence Length | | |
| --- | --- | --- | --- | --- |
| | | Min. | Max. | Average |
| Cytoplasm | 1411 | 9 | 3227 | 337.32 |
| Plasma Membrane | 1238 | 47 | 3678 | 462.21 |
| Extracellular Space | 843 | 22 | 2820 | 194.01 |
| Nucleus | 837 | 16 | 1975 | 341.35 |
| Mitochondrion | 510 | 21 | 991 | 255.78 |
| Chloroplast | 449 | 71 | 1265 | 242.03 |
| Endoplasmic Reticulum | 198 | 79 | 988 | 314.64 |
| Peroxisome | 157 | 21 | 906 | 310.75 |
| Golgi Apparatus | 150 | 116 | 1060 | 300.70 |
| Lysosomal | 103 | 101 | 1744 | 317.81 |
| Vacuole | 63 | 60 | 607 | 297.95 |
| Total | 5959 | - | - | - |

## C.2 CORONAVIRUS HOST DATASET

The unaligned spike protein sequences from various clades of the Coronaviridae family are collected to form this dataset. It contains 5558 spike sequences, which are extracted from GISAID (GISAID Website, 2021) and ViPR (Pickett et al., 2012), accompanied with their respective infected host information. The sequences are distributed among 21 unique hosts. For the classification task, the infected host's name is determined based on a given spike sequence as an input. The statistical detail of this data is shown in Table 9. This data consists of spike protein sequences of the coronavirus and the reason for using only the spike protein region of the virus (rather than the full genome) is because it's well-known that major mutations happen in this region (Kuzmin et al., 2020), possibly due to its ability to attach the virus to the host cell membrane. Figure 6 shows the full genome structure of the SARS-CoV-2 virus and "S" indicates the spike protein region.

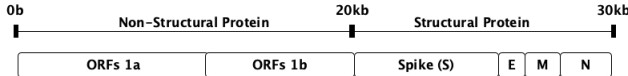

Figure 6: The genome structure of SARS-CoV-2 virus.

Table 9: The distribution of spike protein sequences among the infected hosts. The minimum, maximum, and average lengths of sequences belonging to each host are also mentioned.

| Host | Count | Host Sequence Length | | |
| --- | --- | --- | --- | --- |
| | | Min. | Max. | Average |
| Humans | 1813 | 1154 | 1363 | 1273.63 |
| Environment | 1034 | 19 | 1276 | 1267.56 |
| Weasel | 994 | 9 | 1454 | 1270.69 |
| Swines | 558 | 1130 | 1573 | 1321.34 |
| Birds | 374 | 1153 | 1254 | 1166.93 |
| Camels | 297 | 1169 | 1366 | 1336.39 |
| Bats | 153 | 12 | 1399 | 1292.41 |
| Cats | 123 | 23 | 1467 | 1241.43 |
| Bovines | 88 | 1361 | 1584 | 1375.55 |
| Canis | 40 | 57 | 1473 | 1181.7 |
| Rats | 26 | 1126 | 1378 | 1314.42 |
| Pangolins | 21 | 10 | 1269 | 792.47 |
| Hedgehog | 15 | 1327 | 1363 | 1336.26 |
| Dolphins | 7 | 1482 | 1495 | 1486.85 |
| Equine | 5 | 1363 | 1363 | 1363.00 |
| Fish | 2 | 1190 | 1220 | 1205.0 |
| Python | 2 | 959 | 959 | 959.0 |
| Monkey | 2 | 1273 | 1273 | 1273.0 |
| Unknown | 2 | 1255 | 1271 | 1263.0 |
| Turtle | 1 | 922 | 922 | 922.0 |
| Cattle | 1 | 1169 | 1169 | 1169 |
| Total | 5558 | - | - | - |

## C.3 Anticancer Peptides (ACPs) Dataset

This dataset (Grisoni et al., '2019') consists of unaligned peptide-protein sequences along with their respective anticancer activity on the breast cancer cell lines. We utilize the sequences as inputs and their corresponding anticancer activity as target labels for classification. This data has 949 sequences distributed among the four unique target labels. The statistical detail of this data is given in Table 10.

Table 10: ACPs dataset distribution based on their respective activity on the breast cancer cell line. The minimum, maximum, and average lengths of sequences belonging to each class are also mentioned.

| ACPs Class | Count | Peptide Sequence Length | | |
| | | Min. | Max. | Average |
| --- | --- | --- | --- | --- |
| Inactive-Virtual | 750 | 8 | 30 | 16.64 |
| Moderate Active | 98 | 10 | 38 | 18.44 |
| Inactive-Experimental | 83 | 5 | 38 | 15.02 |
| Very Active | 18 | 13 | 28 | 19.33 |
| Total | 949 | - | - | - |

## C.4 Human DNA Dataset

It consists of 2,000 unaligned Human DNA nucleotide sequences which are distributed among seven unique gene families. These gene families are used as labels for classification. The gene families are G Protein Coupled, Tyrosine Kinase, Tyrosine Phosphatase, Synthetase, Synthase, Ion Channel, and Transcription Factor containing 215, 299, 127, 347, 319, 94, & 599 instances respectively. The statistical detail of this data is given in Table 11.

Table 11: Human DNA dataset distribution based on their gene family type. The minimum, maximum, and average lengths of sequences belonging to each class are also mentioned.

| Gene Family | Count | DNA Sequence Length | | |
| | | Min. | Max. | Average |
| --- | --- | --- | --- | --- |
| G Protein Coupled | 215 | 30 | 18921 | 1859.04 |
| Tyrosine Kinase | 299 | 24 | 4863 | 1509.64 |
| Tyrosine Phosphatase | 127 | 135 | 7473 | 2486.14 |
| Synthetase | 347 | 8 | 3795 | 965.30 |
| Synthase | 319 | 31 | 7536 | 899.63 |
| Ion Channel | 94 | 42 | 5598 | 1949.809 |
| Transcription Factor | 599 | 9 | 9141 | 1152.85 |
| Total | 2000 | - | - | - |

## C.5 SMILES String (Shamay et al., 2018)

It has 6,568 SMILES strings distributed among ten unique drug subtypes extracted from the Drug-Bank dataset. We employ the drug subtypes as a label for classification. The drug subtypes are Barbiturate [EPC], Amide Local Anesthetic [EPC], Non-Standardized Plant Allergenic Extract [EPC], Sulfonylurea [EPC], Corticosteroid [EPC], Nonsteroidal Anti-inflammatory Drug [EPC], Nucleoside Metabolic Inhibitor [EPC], Nitroimidazole Antimicrobial [EPC], Muscle Relaxant [EPC], and Others with 54, 53, 30, 17, 16, 15, 11, 10, 10, & 6,352 instances respectively. The statistical detail of this data is given in Table 12.

## C.6 Music Dataset (Li et al., 2003)

This data has 1,000 audio sequences belonging to 10 unique music genres, where each genre contains 100 sequences. We perform music genre classification task using this dataset. The genres are

Table 12: SMILES String dataset distribution based on their drug subtype. The minimum, maximum, and average lengths of sequences belonging to each class are also mentioned.

| Gene Family | Count | DNA Sequence Length | | |
| | | Min. | Max. | Average |
|---|---|---|---|---|
| Barbiturate [EPC] | 54 | 16 | 136 | 51.24 |
| Amide Local Anesthetic [EPC] | 53 | 9 | 149 | 39.18 |
| Non-Standardized Plant Allergenic Extract [EPC] | 30 | 10 | 255 | 66.89 |
| Sulfonylurea [EPC] | 17 | 22 | 148 | 59.76 |
| Corticosteroid [EPC] | 16 | 57 | 123 | 95.43 |
| Nonsteroidal Anti-inflammatory Drug [EPC] | 15 | 29 | 169 | 53.6 |
| Nucleoside Metabolic Inhibitor [EPC] | 11 | 16 | 145 | 59.90 |
| Nitroimidazole Antimicrobial [EPC] | 10 | 27 | 147 | 103.8 |
| Muscle Relaxant [EPC] | 10 | 9 | 82 | 49.8 |
| Others | 6352 | 2 | 569 | 55.44 |
| Total | 6568 | - | - | - |

Blues, Classical, Country, Disco, Hiphop, Jazz, Metal, Pop, Reggae, and Rock. The statistical detail of this data is given in Table 13

Table 13: Music dataset distribution based on their genre. The minimum, maximum, and average lengths of sequences belonging to each class are also mentioned.

| Genre | Count | Music Sequence Length | | |
| | | Min. | Max. | Average |
|---|---|---|---|---|
| Blues | 100 | 6892 | 6892 | 6892.00 |
| Classical | 100 | 6887 | 7001 | 6895.36 |
| Country | 100 | 6885 | 6974 | 6894.44 |
| Disco | 100 | 6887 | 6958 | 6893.49 |
| Hiphop | 100 | 6889 | 6889 | 6889.00 |
| Jazz | 100 | 6873 | 7038 | 6909.45 |
| Metal | 100 | 6891 | 6999 | 6896.57 |
| Pop | 100 | 6889 | 6892 | 6889.96 |
| Reggae | 100 | 6889 | 6892 | 6890.23 |
| Rock | 100 | 6888 | 6981 | 6894.26 |
| Total | 1000 | - | - | - |

## D   BASELINE MODELS

The details of each of the baseline methods used to perform the evaluation are as follows,

### D.1   ONE-HOT ENCODING (OHE) (KUZMIN ET AL., 2020)

This technique is employed to convert the sequential data into numerical format. For each character within the sequence, a binary vector is associated with it and then all these vectors are combined to form a comprehensive representation of the entire sequence. While OHE is a straightforward approach, it yields vectors that are notably sparse, where most of the elements are zero. This phenomenon is commonly called the "curse of dimensionality," which presents challenges due to the substantial increase in dimensions relative to the available data points.

### D.2   WASSERSTEIN DISTANCE GUIDED REPRESENTATION LEARNING (WDGRL) (SHEN ET AL., 2018)

This domain adaption method is used for adapting data from one domain to another. Its core focus is evaluating the Wasserstein distance (WD) between the encoded distributions of the source and target domains. To achieve it, a neural network is employed. It transforms high-dimensional data into low dimensions. It's worth noting that WDGRL builds upon the feature vectors that are initially generated using the One-Hot Encoding (OHE) technique. However, as it needs large training data to obtain optimal features, therefore it's an expensive mechanism.

FCGR is specifically designed to map protein sequences into images based on the concept of Chaos Game Representation (CGR) (Jeffrey, 1990). For a given protein sequence, it constructs an n-flakes-based image, which consists of multiple icosagons, and $n$ represents the number of amino acids in the sequence. For an amino acid, the corresponding pixel coordinates are determined using the following equations,

$$x = r \cdot sin(\frac{2\pi i}{n} + \theta) \tag{3}$$

$$y = r \cdot cos(\frac{2\pi i}{n} + \theta) \tag{4}$$

where $r$ contraction ratio between the outer and inner polygons.

## D.4   RANDOM CHAOS GAME REPRESENTATION (RANDOMCGR) (MURAD ET AL.)

This method follows a random function to determine the pixel coordinates for an amino acid in the corresponding image representation. These coordinates are further connected with the location axis of the previous amino acid to represent the existing amino acid in the image.

Furthermore, an example of images generated for a sequence taken from the Coronavirus host dataset's bat class against our proposed method and the image-based baseline models is shown in Figure 7. Moreover, the sample images from other datasets are given in Figure 8 Figure 9 Figure 10. We can observe that the constructed images are distinct for each method, which indicates that every image-creating method is able to have different modeling of the information from the sequence in the respective visual form. Hence, every method will achieve different performance in terms of classification.

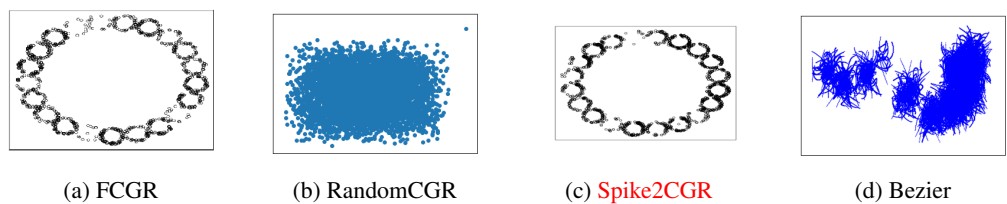

|      (a) FCGR      |    (b) RandomCGR    |    (c) Spike2CGR    |    (d) Bezier    |

Figure 7: The example of images created by our proposed Bézier curve-based method and the image-based baselines methods (FCGR, RandomCGR & Spike2CGR) for a randomly selected sequence from the **Coronavirus host** dataset.

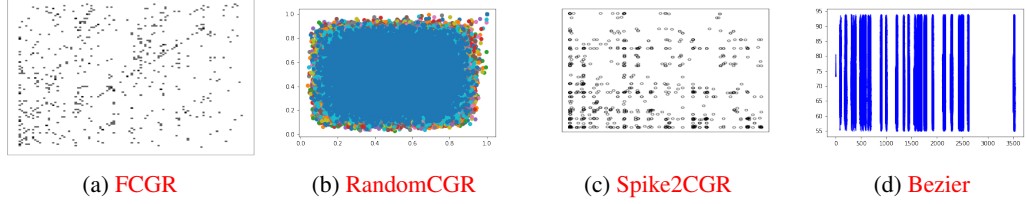

|      (a) FCGR      |    (b) RandomCGR    |    (c) Spike2CGR    |    (d) Bezier    |

Figure 8: The example of images created by our proposed Bézier curve-based method and the image-based baselines methods (FCGR, RandomCGR & Spike2CGR) for a randomly selected sequence from the **Human DNA dataset**.

## E   EXPERIMENTAL EVALUATION

### E.1   EVALUATION METRICS

Various evaluation metrics are employed by us to analyze the performance of the classification models. Those metrics are average accuracy, precision, recall, F1 (weighted), F1 (macro), and ROC

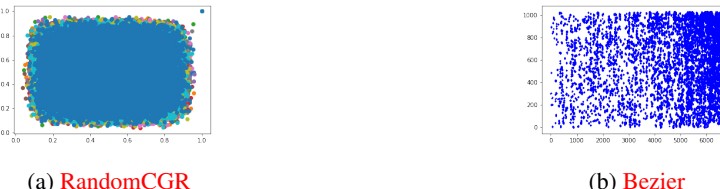

(a) RandomCGR            (b) Bezier

Figure 9: The example of images created by our proposed Bézier curve-based method and the image-based baselines methods (FCGR, RandomCGR & Spike2CGR) for a randomly selected sequence from the **Music dataset**.

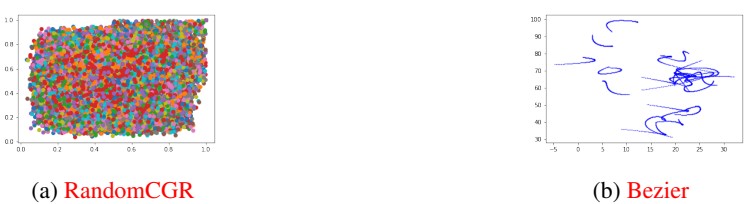

(a) RandomCGR            (b) Bezier

Figure 10: The example of images created by our proposed Bézier curve-based method and the image-based baselines methods (FCGR, RandomCGR & Spike2CGR) for a randomly selected sequence from the **SMILES String dataset**.

AUC. As our classification problems are multi-class, therefore we utilized the one-vs-rest approach for computing the ROC AUC score. The reported values for any metric are an average of 5 runs. Moreover, several metrics are used to obtain deeper insights into the performance of the classifiers, especially because our datasets are undergoing the data imbalance challenge.

### E.1.1 IMAGE MODELS

These models are used for image-based classification. We construct four custom convolutional neural networks (CNNs) classifiers with varying numbers of hidden layers to do the classification tasks. These models are referred to as 1-layer, 2-layer, 3-layer & 4-layer CNN classifiers, and they consist of 1, 2, 3, & 4 hidden block A modules respectively. A block A module contains a convolution layer followed by a ReLu activation function and a max-pool layer. Each convolution layer uses kernel and stride filters of size 5x5 & 2x2, respectively. The max-pooling layer is also accompanied by the kernel and stride filters of 2x2 sizes for both. For each classifier, the block A modules are followed by two fully connected layers and a Softmax classification layer. The architecture of the 1-layer CNN model is illustrated in Figure 11. These custom CNN networks are employed to investigate the impact of increasing the number of hidden layers on the final predictive performance.

Moreover, a vision transformer model (ViT) is also used by us for performing the classification tasks. As ViT is known to utilize the power of transformer architecture, we want to see its impact on our bio-sequence datasets classifications. In ViT the input image is partitioned into patches, which are then linearly transformed into vectors by a linear embedding module. Note that we used patch size 20 & 8 vector dimensions in our experiments. Then positional embeddings are added to the vectors and they are subsequently processed by two Transformer encoder blocks. Each encoder block consists of a normalization layer, a multi-head self-attention layer with residual connections, a second normalization layer, and a multi-layer perceptron with another residual connection. The final output is directed to a softmax classification module for image label prediction. This design capitalizes on self-attention mechanisms for efficient image classification.

Furthermore, we also examine the consequences of using pre-trained vision models for classifying our datasets, and for that, we used pre-trained ResNet-50 (He et al., 2016), EfficientNet (Tan & Le, 2019), DenseNet (Iandola et al., 2014) and VGG19 (Simonyan & Zisserman, 2015) models. Moreover, all the generated images for any dataset are of dimensions 720x480, and they are given as input to the image-based classifiers. The hyperparameters decided after fine-tuning are 0.003 learn-

ing rate, ADAM optimizer, 64 batch size, and 10 training epochs for all the models. Additionally, the negative log-likelihood (NLL) (Yao et al., 2019) loss function is employed for training, as it's known to be a cross-entropy loss function for multi-class problems.

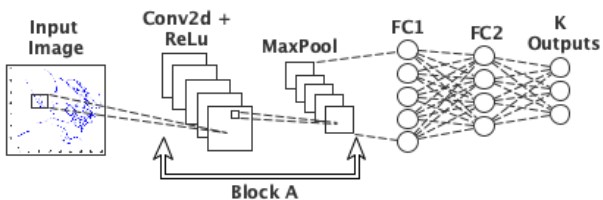

Figure 11: The architecture of 1-layer CNN model. For a given input image, it yields the K classification classes as output.

## F   RESULTS AND DISCUSSION

### F.1   CORONAVIRUS HOST DATASET'S PERFORMANCE

The Coronavirus host dataset-based classification performance via various evaluation metrics is reported in Table 14.

The results demonstrate that overall the Bezier-based images are showing promising performance for the host classification task. Among the image-based baselines, it portrays a very comparable performance with the FCGR baseline and has outperformed the RandomCGR method. Moreover, the image representations are clearly performing better than the tabular ones (feature-engineering methods based). Hence converting the spike protein sequences of the Coronavirus into images is more effective for performing classification as compared to transforming them into numerical vectors (tabular form).

### F.2   ACP DATASET'S PERFORMANCE

The classification performance achieved using the ACP dataset for various evaluation metrics is summarized in Table 15.

We can observe that, among the image-based baseline approaches Bezier is showcasing very comparable performance as the FCGR because most of the top 5% scores corresponding to various evaluation metrics are falling against these methods. Moreover, Bezier is clearly outperforming the RandomCGR baseline. Although the OHE method exhibits high performance for some of the evaluation metrics, generally the feature engineering methods have lower predictive performance than the image-based ones. Overall, we can say that the Bezier method portrays promising results for the ACP classification task, which implies that converting the ACP sequences into images using the Bezier method is effective in terms of classification performance.

### F.3   T-SNE DATA VISUALIZATION

The t-SNE plots for the Coronavirus host dataset are shown in Figure 12. We can observe that, the clusters generated by both FCGR and Bézier are very scattered and overlapping. This is an implication that the embeddings created by both these methods are almost similar in quality as they possess somehow similar data preservation patterns in a low-dimensional space.

Furthermore, the t-SNE visualization of the ACP dataset is given in Figure 13. It also portrays a very similar cluster structure for both FCGR and Bézier methods. The clusters are very scattered, non-definite, and overlapping, which indicates that the respective embeddings are unable to preserve the data structure in 2D space.

The t-SNE plots of the Human DNA dataset are given in Figure 14. We can observe that the Bezier-based clusters are more compact while FCGR ones are very scattered.

Table 14: Classification results for different models and algorithms for **Coronavirus Host dataset**. The top 5% values for each metric are underlined.

| Category | DL Model | Method | Acc. ↑ | Prec. ↑ | Recall ↑ | F1 (Weig.) ↑ | F1 (Macro) ↑ | ROC AUC ↑ | Train Time (hrs.) ↓ |
|---|---|---|---|---|---|---|---|---|---|
| Tabular Models | 3-Layer Tab CNN | OHE | 0.625 | 0.626 | 0.625 | 0.566 | 0.335 | 0.663 | 0.032 |
| | | WDGRL | 0.304 | 0.137 | 0.304 | 0.182 | 0.041 | 0.499 | 0.029 |
| | 4-Layer Tab CNN | OHE | 0.613 | 0.478 | 0.613 | 0.534 | 0.323 | 0.662 | 0.067 |
| | | WDGRL | 0.312 | 0.130 | 0.312 | 0.167 | 0.035 | 0.498 | 0.054 |
| String Kernel | - | SVM | 0.601 | 0.673 | 0.601 | 0.602 | 0.325 | 0.624 | 5.198 |
| | - | NB | 0.230 | 0.665 | 0.230 | 0.295 | 0.162 | 0.625 | 0.131 |
| | - | MLP | 0.647 | 0.696 | 0.647 | 0.641 | 0.302 | 0.628 | 42.322 |
| | - | KNN | 0.613 | 0.623 | 0.613 | 0.612 | 0.310 | 0.629 | 0.434 |
| | - | RF | 0.668 | 0.692 | 0.668 | 0.663 | 0.360 | 0.658 | 4.541 |
| | - | LR | 0.554 | 0.724 | 0.554 | 0.505 | 0.193 | 0.568 | 5.096 |
| | - | DT | 0.646 | 0.674 | 0.646 | 0.643 | 0.345 | 0.653 | 1.561 |
| Custom CNN Models | 1-Layer | FCGR | 0.680 | 0.707 | 0.680 | 0.670 | 0.517 | 0.761 | 0.984 |
| | | Spike2CGR | 0.743 | 0.745 | 0.743 | 0.739 | 0.569 | 0.797 | 0.711 |
| | | RandomCGR | 0.262 | 0.193 | 0.262 | 0.210 | 0.051 | 0.500 | 8.695 |
| | | Bézier | 0.652 | 0.652 | 0.652 | 0.644 | 0.592 | 0.766 | 2.698 |
| | % improv. of Bézier from FCGR | | -2.8 | -5.5 | -2.8 | -2.6 | 7.5 | 0.5 | -174.18 |
| | 2-Layer | FCGR | 0.668 | 0.684 | 0.668 | 0.655 | 0.410 | 0.710 | 1.046 |
| | | Spike2CGR | 0.740 | 0.734 | 0.740 | 0.726 | 0.428 | 0.716 | 0.688 |
| | | RandomCGR | 0.293 | 0.235 | 0.293 | 0.246 | 0.093 | 0.521 | 8.839 |
| | | Bézier | 0.656 | 0.669 | 0.656 | 0.644 | 0.610 | 0.778 | 2.976 |
| | % improv. of Bézier from FCGR | | -1.2 | -1.5 | -1.2 | -1.1 | 20 | 6.8 | -184.51 |
| | 3-Layer | FCGR | 0.681 | 0.677 | 0.681 | 0.672 | 0.470 | 0.740 | 5.681 |
| | | Spike2CGR | 0.729 | 0.729 | 0.729 | 0.715 | 0.354 | 0.677 | 0.831 |
| | | RandomCGR | 0.320 | 0.102 | 0.320 | 0.155 | 0.028 | 0.500 | 9.440 |
| | | Bézier | 0.611 | 0.652 | 0.611 | 0.612 | 0.623 | 0.793 | 4.660 |
| | % improv. of Bézier from FCGR | | -7 | -2.5 | -7 | -6 | 15.3 | 5.3 | 17.97 |
| | 4-Layer | FCGR | 0.624 | 0.617 | 0.624 | 0.606 | 0.262 | 0.623 | 8.991 |
| | | Spike2CGR | 0.686 | 0.668 | 0.686 | 0.672 | 0.283 | 0.632 | 0.684 |
| | | RandomCGR | 0.320 | 0.102 | 0.320 | 0.155 | 0.028 | 0.500 | 10.778 |
| | | Bézier | 0.640 | 0.643 | 0.640 | 0.575 | 0.594 | 0.782 | 5.102 |
| | % improv. of Bézier from FCGR | | 1.6 | 2.6 | 1.6 | -3.1 | 33.2 | 15.9 | 43.25 |
| Vision Transformer | ViT | FCGR | 0.322 | 0.104 | 0.322 | 0.157 | 0.023 | 0.500 | 0.188 |
| | | Spike2CGR | 0.332 | 0.323 | 0.332 | 0.333 | 0.213 | 0.500 | 0.877 |
| | | RandomCGR | 0.320 | 0.102 | 0.320 | 0.155 | 0.028 | 0.500 | 0.173 |
| | | Bézier | 0.316 | 0.100 | 0.316 | 0.152 | 0.022 | 0.500 | 0.183 |
| | % improv. of Bézier from FCGR | | -0.6 | -0.4 | -0.6 | -0.5 | -0.1 | 0 | 2.65 |
| Pretrained Vision Models | ResNet-50 | FCGR | 0.662 | 0.665 | 0.662 | 0.639 | 0.267 | 0.621 | 8.840 |
| | | Spike2CGR | 0.691 | 0.683 | 0.691 | 0.663 | 0.270 | 0.624 | 0.786 |
| | | RandomCGR | 0.319 | 0.113 | 0.319 | 0.159 | 0.030 | 0.500 | 13.488 |
| | | Bézier | 0.571 | 0.473 | 0.571 | 0.504 | 0.335 | 0.564 | 6.411 |
| | % improv. of Bézier from FCGR | | -9.1 | -19.2 | -9.1 | -13.5 | 6.8 | -5.7 | 27.47 |
| | VGG-19 | FCGR | 0.519 | 0.475 | 0.519 | 0.442 | 0.158 | 0.572 | 3.738 |
| | | Spike2CGR | 0.458 | 0.409 | 0.458 | 0.363 | 0.129 | 0.559 | 3.409 |
| | | RandomCGR | 0.320 | 0.102 | 0.320 | 0.155 | 0.028 | 0.500 | 21.474 |
| | | Bézier | 0.521 | 0.421 | 0.521 | 0.448 | 0.222 | 0.500 | 3.200 |
| | % improv. of Bézier from FCGR | | 0.2 | -5.4 | 0.2 | 0.6 | 6.4 | -7.2 | 14.39 |
| | DenseNet | FCGR | 0.018 | 0.000 | 0.018 | 0.001 | 0.018 | 0.500 | 2.566 |
| | | Spike2CGR | 0.017 | 0.000 | 0.017 | 0.000 | 0.001 | 0.500 | 2.675 |
| | | RandomCGR | 0.015 | 0.000 | 0.015 | 0.000 | 0.001 | 0.500 | 2.123 |
| | | Bézier | 0.011 | 0.000 | 0.011 | 0.001 | 0.011 | 0.500 | 2.332 |
| | % improv. of Bézier from FCGR | | -0.8 | 0 | -0.8 | 0 | -0.8 | 0 | 9.11 |
| | EfficientNet | FCGR | 0.169 | 0.028 | 0.169 | 0.049 | 0.013 | 0.500 | 34.443 |
| | | Spike2CGR | 0.169 | 0.031 | 0.169 | 0.053 | 0.015 | 0.500 | 31.229 |
| | | RandomCGR | 0.317 | 0.108 | 0.317 | 0.162 | 0.032 | 0.529 | 37.334 |
| | | Bézier | 0.465 | 0.427 | 0.465 | 0.394 | 0.157 | 0.577 | 35.768 |
| | % improv. of Bézier from FCGR | | 29.6 | 39.9 | 29.6 | 34.5 | 14.4 | 7.7 | -3.84 |

## F.4 INTER-CLASS CORRELATION

To analyze the correlation between different classes of our respective datasets we utilized the heat maps. These maps are generated from the embeddings extracted from the last layer of the 2-layer CNN classifier corresponding to FCGR and Bézier based images respectively. To construct the maps, first pairwise cosine similarity scores are computed between embeddings of different classes

Table 15: Classification results for different models and algorithms for **ACPs (Breast Cancer) dataset**. The top 5% values for each metric are underlined.

| Category | DL Model | Method | Acc. ↑ | Prec. ↑ | Recall ↑ | F1 (Weig.) ↑ | F1 (Macro) ↑ | ROC AUC ↑ | Train Time (hrs.) ↓ |
|---|---|---|---|---|---|---|---|---|---|
| Tabular Models | 3-Layer Tab CNN | OHE | 0.768 | 0.839 | 0.768 | 0.790 | 0.452 | 0.719 | 0.042 |
| | | WDGRL | 0.615 | 0.740 | 0.615 | 0.660 | 0.326 | 0.603 | 0.0001 |
| | 4-Layer Tab CNN | OHE | 0.796 | 0.843 | 0.796 | 0.807 | 0.474 | 0.736 | 0.056 |
| | | WDGRL | 0.631 | 0.754 | 0.631 | 0.673 | 0.346 | 0.623 | 0.0002 |
| String Kernel | - | SVM | 0.802 | 0.836 | 0.802 | 0.813 | 0.454 | 0.692 | 0.789 |
| | - | NB | 0.872 | 0.869 | 0.872 | 0.864 | 0.523 | 0.732 | 0.018 |
| | - | MLP | 0.611 | 0.771 | 0.611 | 0.666 | 0.348 | 0.626 | 2.478 |
| | - | KNN | 0.871 | 0.849 | 0.871 | 0.853 | 0.482 | 0.694 | 0.286 |
| | - | RF | 0.866 | 0.837 | 0.866 | 0.847 | 0.470 | 0.681 | 1.029 |
| | - | LR | 0.881 | 0.872 | 0.881 | 0.870 | 0.536 | 0.720 | 0.254 |
| | - | DT | 0.835 | 0.843 | 0.835 | 0.838 | 0.465 | 0.702 | 0.338 |
| Custom CNN Models | 1-Layer | FCGR | 0.863 | 0.831 | 0.863 | 0.844 | 0.490 | 0.677 | 0.357 |
| | | Spike2CGR | 0.783 | 0.613 | 0.783 | 0.687 | 0.219 | 0.500 | 0.999 |
| | | RandomCGR | 0.792 | 0.638 | 0.792 | 0.707 | 0.221 | 0.497 | 0.404 |
| | | Bézier | 0.835 | 0.779 | 0.835 | 0.781 | 0.314 | 0.548 | 0.805 |
| | % improv. of Bézier from FCGR | | -2.8 | -5.2 | -2.8 | -6.3 | -17.6 | -12.9 | -125.49 |
| | 2-Layer | FCGR | 0.852 | 0.833 | 0.852 | 0.837 | 0.489 | 0.676 | 0.419 |
| | | Spike2CGR | 0.783 | 0.613 | 0.783 | 0.687 | 0.219 | 0.500 | 1.196 |
| | | RandomCGR | 0.800 | 0.640 | 0.800 | 0.711 | 0.222 | 0.500 | 0.389 |
| | | Bézier | 0.814 | 0.795 | 0.814 | 0.803 | 0.419 | 0.633 | 0.626 |
| | % improv. of Bézier from FCGR | | -3.8 | -3.8 | -3.8 | -3.4 | -7 | -4.3 | -49.40 |
| | 3-Layer | FCGR | 0.800 | 0.640 | 0.800 | 0.711 | 0.222 | 0.500 | 0.490 |
| | | Spike2CGR | 0.783 | 0.612 | 0.783 | 0.687 | 0.219 | 0.500 | 1.456 |
| | | RandomCGR | 0.800 | 0.640 | 0.800 | 0.711 | 0.222 | 0.500 | 0.391 |
| | | Bézier | 0.830 | 0.748 | 0.830 | 0.780 | 0.296 | 0.541 | 0.637 |
| | % improv. of Bézier from FCGR | | 3 | 10.8 | 3 | 6.9 | 7.4 | 4.1 | -30 |
| | 4-Layer | FCGR | 0.831 | 0.735 | 0.831 | 0.779 | 0.329 | 0.586 | 0.498 |
| | | Spike2CGR | 0.783 | 0.612 | 0.783 | 0.687 | 0.219 | 0.500 | 1.776 |
| | | RandomCGR | 0.800 | 0.640 | 0.800 | 0.711 | 0.222 | 0.500 | 0.435 |
| | | Bézier | 0.825 | 0.681 | 0.825 | 0.746 | 0.226 | 0.500 | 0.668 |
| | % improv. of Bézier from FCGR | | -0.6 | -5.4 | -0.6 | -3.3 | -10.3 | -8.6 | -34.13 |
| Vision Transformer | ViT | FCGR | 0.767 | 0.588 | 0.767 | 0.666 | 0.217 | 0.500 | 0.031 |
| | | Spike2CGR | 0.754 | 0.487 | 0.74 | 0.565 | 0.211 | 0.500 | 0.650 |
| | | RandomCGR | 0.756 | 0.512 | 0.756 | 0.632 | 0.201 | 0.500 | 0.032 |
| | | Bézier | 0.825 | 0.681 | 0.825 | 0.746 | 0.226 | 0.500 | 0.027 |
| | % improv. of Bézier from FCGR | | 5.8 | 9.3 | 5.8 | 8 | 0.9 | 0 | 12.90 |
| Pretrained Vision Models | ResNet-50 | FCGR | 0.800 | 0.642 | 0.800 | 0.712 | 0.222 | 0.501 | 1.317 |
| | | Spike2CGR | 0.770 | 0.559 | 0.770 | 0.654 | 0.198 | 0.500 | 2.290 |
| | | RandomCGR | 0.800 | 0.640 | 0.800 | 0.711 | 0.222 | 0.500 | 1.387 |
| | | Bézier | 0.835 | 0.780 | 0.835 | 0.796 | 0.334 | 0.601 | 0.175 |
| | % improv. of Bézier from FCGR | | 3.5 | 13.8 | 3.5 | 8.4 | 11.2 | 10 | 86.71 |
| | VGG-19 | FCGR | 0.803 | 0.684 | 0.803 | 0.720 | 0.243 | 0.509 | 1.189 |
| | | Spike2CGR | 0.765 | 0.650 | 0.765 | 0.650 | 0.200 | 0.500 | 2.111 |
| | | RandomCGR | 0.800 | 0.640 | 0.800 | 0.711 | 0.222 | 0.500 | 1.054 |
| | | Bézier | 0.825 | 0.681 | 0.825 | 0.746 | 0.226 | 0.500 | 2.144 |
| | % improv. of Bézier from FCGR | | 2.2 | -0.3 | 2.2 | 2.6 | -1.7 | -0.9 | -80.31 |
| | DenseNet | FCGR | 0.116 | 0.013 | 0.116 | 0.024 | 0.052 | 0.500 | 0.987 |
| | | Spike2CGR | 0.116 | 0.011 | 0.116 | 0.022 | 0.050 | 0.500 | 1.767 |
| | | RandomCGR | 0.095 | 0.011 | 0.095 | 0.010 | 0.095 | 0.500 | 1.381 |
| | | Bézier | 0.105 | 0.011 | 0.105 | 0.020 | 0.105 | 0.500 | 1.211 |
| | % improv. of Bézier from FCGR | | -1.1 | -0.2 | -1.1 | -0.4 | 5.3 | 0 | -22.69 |
| | EfficientNet | FCGR | 0.089 | 0.008 | 0.089 | 0.014 | 0.041 | 0.500 | 1.622 |
| | | Spike2CGR | 0.085 | 0.005 | 0.085 | 0.009 | 0.008 | 0.500 | 2.221 |
| | | RandomCGR | 0.028 | 0.002 | 0.028 | 0.004 | 0.027 | 0.500 | 1.988 |
| | | Bézier | 0.058 | 0.003 | 0.058 | 0.006 | 0.027 | 0.500 | 1.566 |
| | % improv. of Bézier from FCGR | | -3.1 | -0.5 | -3.1 | -0.8 | -1.4 | 0 | 3.45 |

and then an average value is calculated to get a score for a class. Note that the maps are normalized between [0-1] to the identity pattern.

The heat maps of the protein subcellular dataset corresponding to FCGR and our proposed method are demonstrated in Figure 15. We can observe that in the case of FCGR, although each class has a maximum correlation with itself, some of them also portray a high correlation with each other as well. For instance, the "Golgi" class shows a high similarity to "ER" and "Pero" etc. This

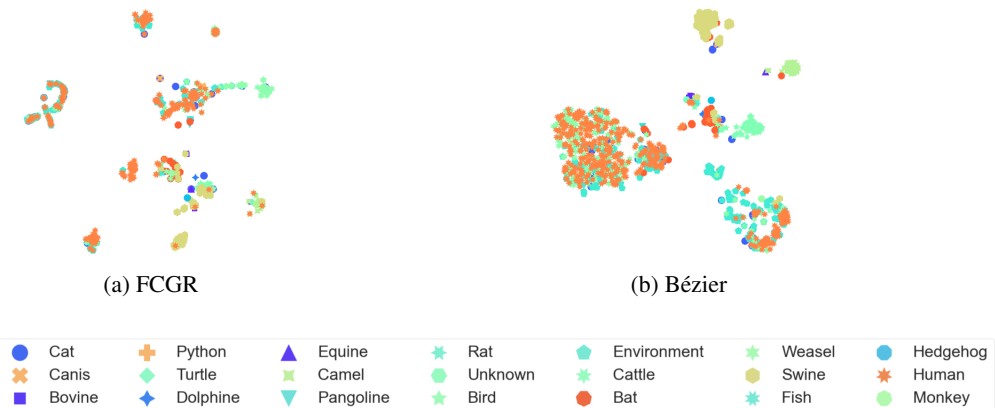

(a) FCGR      (b) Bézier

| | | | | | | |
|---|---|---|---|---|---|---|
| ● Cat | ✚ Python | ▲ Equine | ✳ Rat | ⬠ Environment | ✳ Weasel | ● Hedgehog |
| ✖ Canis | ◆ Turtle | ✖ Camel | ● Unknown | ✳ Cattle | ● Swine | ✳ Human |
| ■ Bovine | ✦ Dolphine | ▼ Pangoline | ★ Bird | ● Bat | ✳ Fish | ● Monkey |

Figure 12: The t-SNE plots of **Coronavirus Host dataset** embeddings extracted from the last layer of **2layer CNN** classifier using the FCGR- and Bézier-based images respectively. The figure is best seen in color.

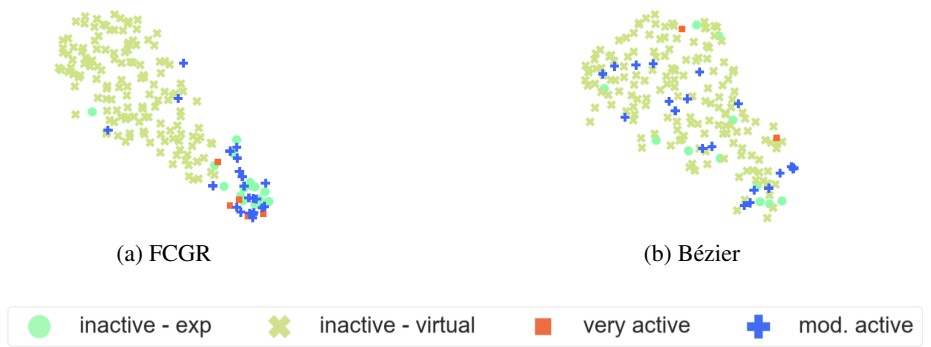

(a) FCGR      (b) Bézier

| | | | |
|---|---|---|---|
| ● inactive - exp | ✖ inactive - virtual | ■ very active | ✚ mod. active |

Figure 13: The t-SNE plots of **ACP dataset** embeddings extracted from the last layer of **2layer CNN** classifier using the FCGR- and Bézier-based images respectively. The figure is best seen in color.

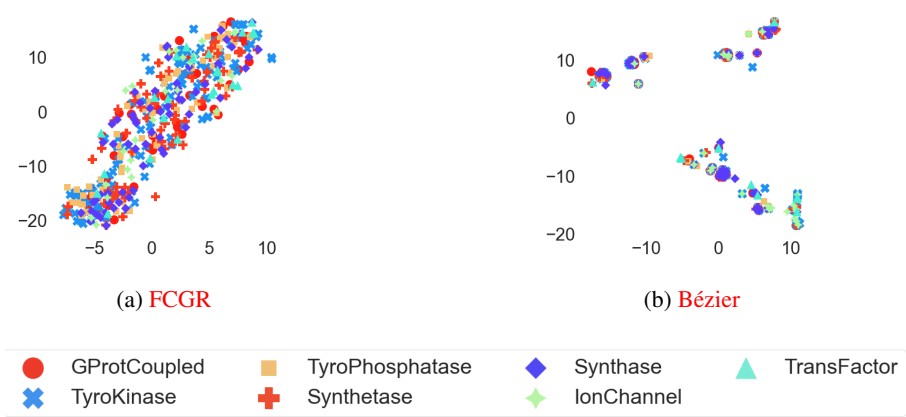

(a) FCGR      (b) Bézier

| | | | |
|---|---|---|---|
| ● GProtCoupled | ■ TyroPhosphatase | ◆ Synthase | ▲ TransFactor |
| ✖ TyroKinase | ✚ Synthetase | ◆ IonChannel | |

Figure 14: The t-SNE plots of **Human dataset** embeddings extracted from the last layer of **2layer CNN** classifier using the FCGR- and Bézier-based images respectively. The figure is best seen in color.

indicates that distinguishing different classes is hard using the embeddings generated by FCGR, hence the FCGR images are suboptimal representations. However, the heat map constructed from

the Bézier images shows that each class has maximum similarity to itself only and holds almost no correlation with other classes. It is an implication that the embeddings generated from the Bézier method belonging to the same class are very similar to each other, while highly distinct from the embeddings of other classes. Hence, the Bézier-based images are optimal and it can also be proven by the classification performance achieved by them for our respective dataset.

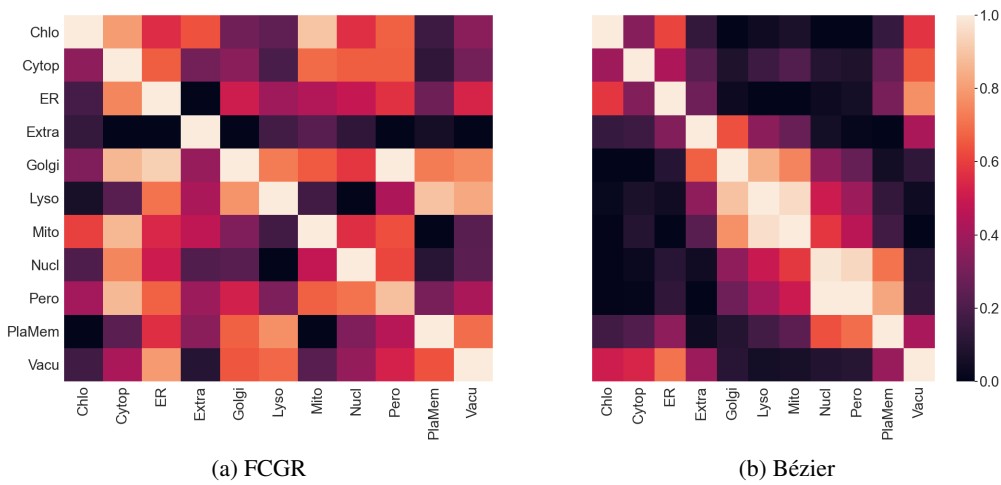

|                | (a) FCGR | (b) Bézier |
| -------------- | -------- | ---------- |

Figure 15: Heatmap for cosine similarity between different **Protein Subcellular Localization** pairs for FCGR and Bézier-based image generation methods corresponding to 2layer CNN classifier.

Similarly, the heat maps for cosine similarity of the Coronavirus host dataset corresponding to FCGR and Bézier encoding methods are shown in Figure 16. We can observe that for both methods, each class has maximum similarity to itself. However, some of the classes also portray a correlation with other classes, like "Weasel" is highly correlated to "cat" & "can" classes, etc. This means that although the embeddings generated by both FCGR and Bézier which belong to the same class are very similar to each other, they can also be similar to embeddings from some of the other classes.

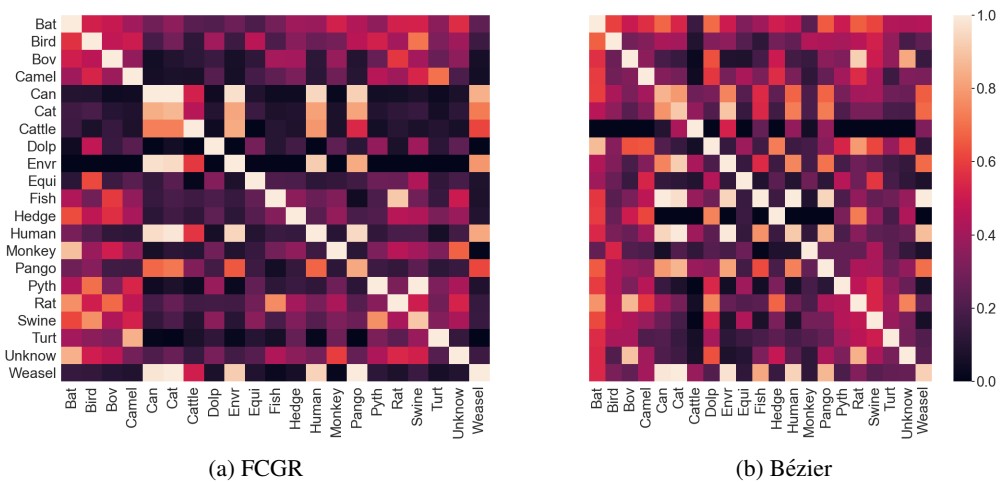

|                | (a) FCGR | (b) Bézier |
| -------------- | -------- | ---------- |

Figure 16: Heatmap for cosine similarity between different **Coronavirus Host** pairs for FCGR and Bézier-based image generation methods corresponding to 2layer CNN classifier.

Furthermore, the heat maps corresponding to the Human DNA dataset are given in Figure 17. We can observe that, in all the maps each class shows maximum similarity to itself.

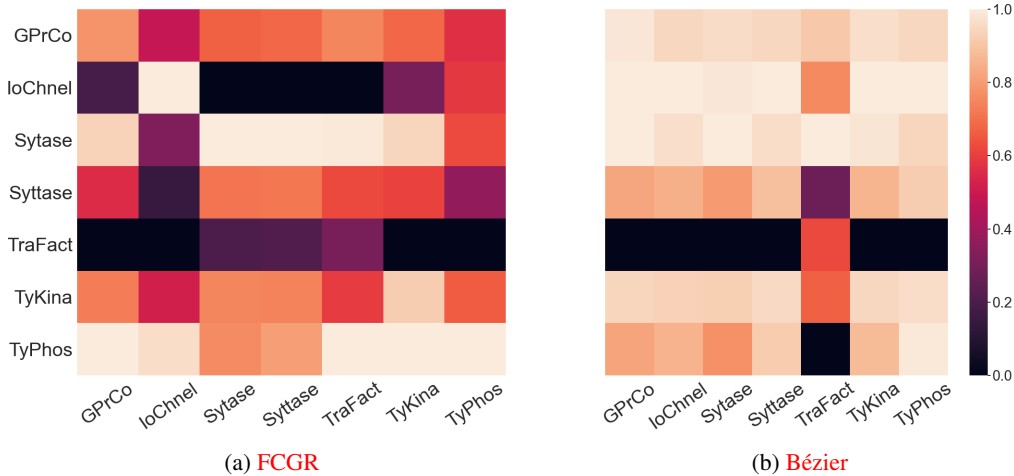

(a) FCGR  (b) Bézier

Figure 17: Heatmap for cosine similarity between different **Human DNA** pairs for FCGR and Bézier-based image generation methods corresponding to 2layer CNN classifier.

## F.5 INTERPRETABLITY

To further evaluate the effectiveness of the created images by FCGR and Bézier methods, we plotted the histograms of the respective embeddings extracted from the last layer of the 2-layer CNN model against the protein subcellular dataset. We employ the 2-layer CNN model because it demonstrates good classification performance for both FCGR and Bézier methods.

The Bézier images-based histograms of the protein subcellular dataset are shown in Figure 18. We can observe that, although the embeddings of (a) & (b) belong to the same label Lysosomal, they yield different histograms which indicate that the information in the respective images is captured differently for different sequences by our Bézier method. Moreover, as they belong to the same category, the Euclidean distance between them is small (0.17), while the distance for embeddings from different categories is large as shown in (c) & (d) which is 0.87. This implies that our method is effective as it keeps similar instances close to each other while the different ones are far from each other.

Similarly, the histograms for FCGR images-based embeddings are illustrated in Figure 19. We can see that the Euclidean distance between the embeddings from the same Lysosomal category ((a) & (b)) is very large (0.88) and it's not a desirable behavior. This indicates that the images generated by FCGR are suboptimal representations of the sequences.

## F.6 CONFUSION MATRIX RESULTS AND DISCUSSION

The confusion matrices for the host dataset are given in Figure 20. We can observe that although FCGR has a high number of true positive values for most of the classes, our method also portrays comparable results. Note that our technique has a high true positive count for the human class, which is the most frequent class in the dataset, as compared to the FCGR method.

Similarly, the confusion matrices for the Human DNA dataset are illustrated in Figure 21, and they portray similar patterns as the host dataset's matrices.

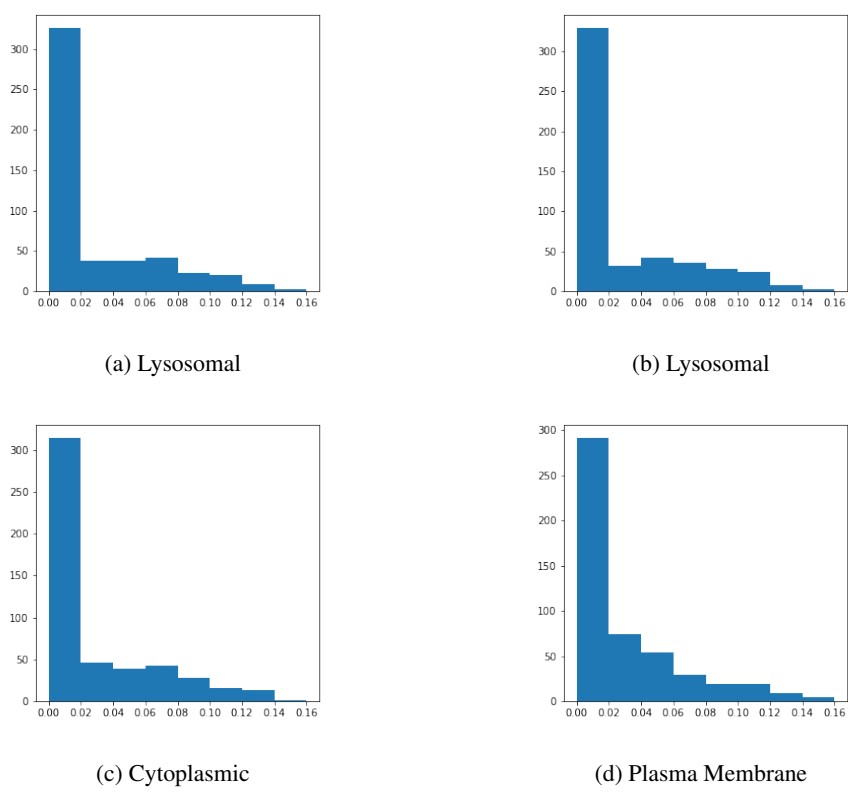

(a) Lysosomal

(b) Lysosomal

(c) Cytoplasmic

(d) Plasma Membrane

Figure 18: The histogram of embeddings extracted from the last layer of the 2-layer CNN model for the **Bézier**-based images of **Protein Subcellular dataset**. (a) & (b) shows the plots for two different embeddings belonging to the **Lysosomal** class and they have a **0.17** Euclidean distance. (c) & (d) contain the histograms of embeddings belonging to **Cytoplasmic** and **Plasma Membrane** classes respectively, and they have an Euclidean distance of **0.87**.

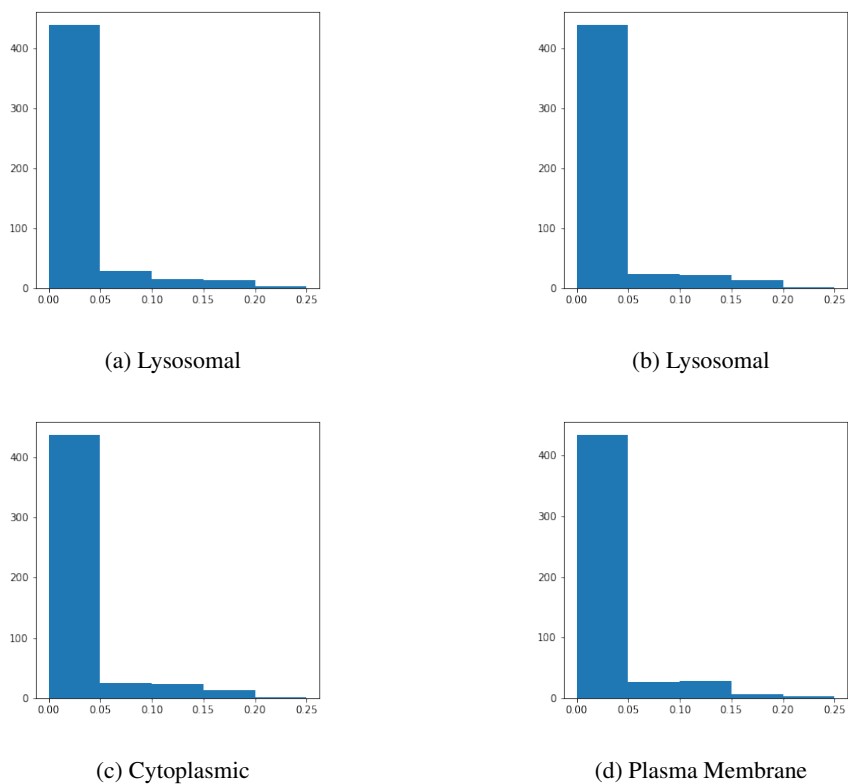

(a) Lysosomal

(b) Lysosomal

(c) Cytoplasmic

(d) Plasma Membrane

Figure 19: The histogram of embeddings extracted from the last layer of the 2-layer CNN model for the **FCGR**-based images of **Protein Subcellular dataset**. (a) & (b) shows the plots for two different embeddings belonging to the **Lysosomal** class and they have a **0.88** Euclidean distance. (c) & (d) contain the histograms of embeddings belonging to **Cytoplasmic** and **Plasma Membrane** classes respectively, and they have an Euclidean distance of **1.30**.

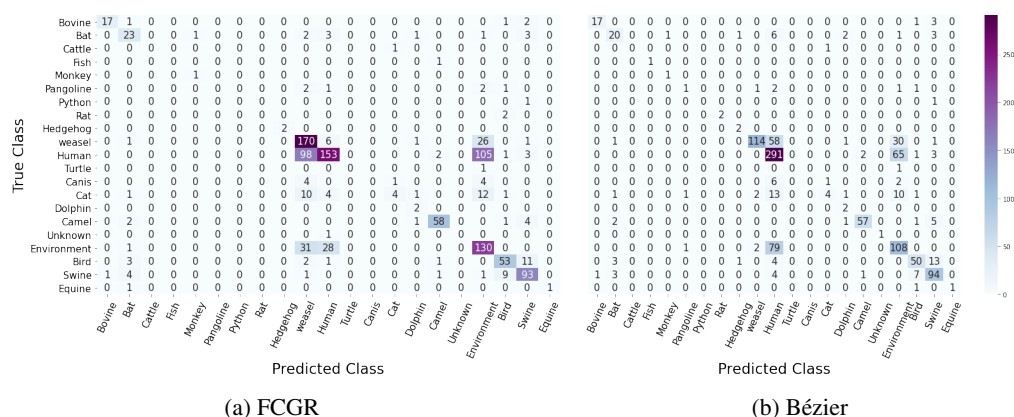

(a) FCGR

(b) Bézier

Figure 20: Confusion matrices of **Coronavirus host dataset** for **2layer CNN classifier** using the FCGR- and Bézier-based image generation methods.

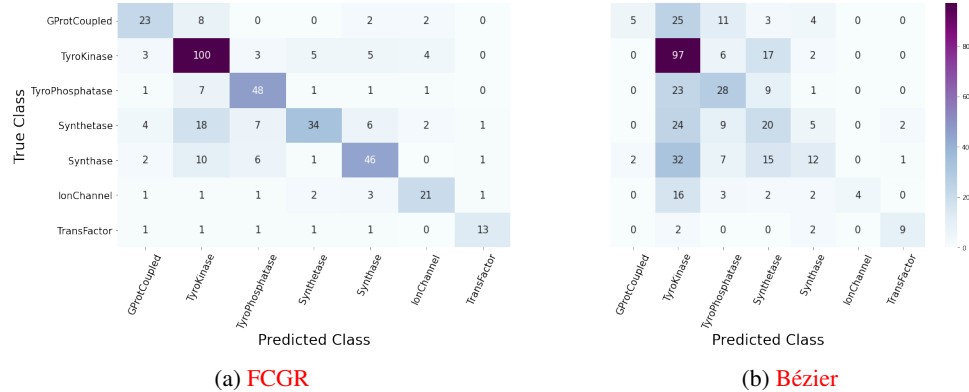

(a) FCGR             (b) Bézier

Figure 21: Confusion matrices of **Human DNA dataset** for **2layer CNN classifier** using the FCGR- and Bézier-based image generation methods.

