# OpenReview forum: "Biological Sequence Analysis Using B ́ezier Curve"
_ICLR.cc/2024/Conference — Submitted to ICLR 2024_

### Official Review · Reviewer_TS9L · 2023-10-31

**Soundness:** 3 good
**Presentation:** 3 good
**Contribution:** 3 good
**Rating:** 6
**Confidence:** 3

**Summary:**

The authors describe an algorithm for representing sequences of amino acids as a collection of Bezier Curves in the plane, and show that using these as inputs for classification tasks is superior to using other 1D and 2D representations of the sequence.

**Strengths:**

- well written
- extensive exploration of different representations of their Bezier curve method

**Weaknesses:**

- For a paper about 2D images, there are very few images. The algorithm is complicated enough to warrant at least one example where the authors illustrate the creation an image from a simple sequence. Moreover, they could compare this to other image production methods and demonstrate why they think their images are superior.
- Train time is a rather weak method for understanding the compute requirements of each method, it would be better to report training flops, and even scatter plot by training flops.

**Questions:**

- What about a transformer on the sequence itself, rather than just a ViT on images? Seems like an important comparison to make.

---

> ### Author Response · Authors · 2023-11-20
> **Authors Response to the Reviewers Comments**
>
> We thank the reviewer for appreciating our work and giving us valuable feedback. Please find below our response to the reviewers comments
>
> ```
> For a paper about 2D images, there are very few images. The algorithm is complicated enough to warrant at least one example where the authors illustrate the creation an image from a simple sequence. Moreover, they could compare this to other image production methods and demonstrate why they think their images are superior.
> ```
>
> We agree with the reviewer for the inclusion of more images. The main reason for not including more images was the page size issue during the original submission. In the revised version of the paper, we have now included an image for the Spike2CGR baseline in Figure 7 in the appendix. Similarly, we also include the images for different methods in Figures 8, 9, and 10 for the Human DNA, Music, and SMILES string datasets, respectively (in the appendix). We will further include more images in the camera-ready version including a visualization showing how the image is created from start to end. Since such an interactive image would not work in the PDF format, we will include that on our website, which we cannot do right now to maintain anonymity.
>
> ```
> Train time is a rather weak method for understanding the compute requirements of each method, it would be better to report training flops, and even scatter plot by training flops.
> ```
>
> The training time is just to get an idea regarding how the vision model behaves for different types of images as input along with how they behave compared to each other in terms of runtime vs. classification results. Since the vision models are just used for downstream task and are not part of our image representation learning idea, that is why we did not included training flops for those models.
>
> ```
> What about a transformer on the sequence itself, rather than just a ViT on images? Seems like an important comparison to make.
> ```
>
> We want to mention that we tried the ``ProteinBERT`` model on the sequences but the results were not very encouraging compared to the baseline results shown in the paper. Therefore, we decided to use other models as shown in the paper. Moreover, the transformer architectures are domain-specific. Since our idea can be generalized to different types of sequence data, we believe that such a comparison would not be that important. Also, the main focus of this paper is to observe the performance of vision models in terms of sequence classification. Therefore, considering all scenarios, we decided to not include transformer models that take sequences as input.
>
>
> References:
>
> [1] Brandes N, Ofer D, Peleg Y, Rappoport N, Linial M. ProteinBERT: a universal deep-learning model of protein sequence and function. Bioinformatics. 2022 Apr 15;38(8):2102-10.

---

> > ### Author Response · Authors · 2023-11-23
> > **Authors Response Regarding Interactive Plot Comment From the Reviewer**
> >
> > Dear Reviewer, to show the interactive plot demo, we have added a sample Jupyter notebook code into the supplementary material (i.e. Bezier_Interactive_Plot jupyter notebook file), which, if you run, will generate a kind of interactive plot where you can visually see how the bezier curve-based plot is generated for each amino acid within a randomly generated protein sequence.
> >
> > Due to the annonimity, we show the interactive plot for a sample sequence rather than actual data as it is easy to run just to get an idea of how our method works. We hope that this will help you understand our idea better.

---

> ### Author Response · Authors · 2023-11-21
> **Second Response To The Reviewer's Comments**
>
> Dear reviewer, as the end of the discussion period is fast approaching, please do let us know if you have any further suggestions or concerns to which we can respond. We'd be interested to hear if our response and changes to the manuscript helped answer your questions. If you are satisfied with our response, we would kindly request you to increase your score if possible, so that we have more chances of paper acceptance.

---

> > ### Comment · Reviewer_TS9L · 2023-11-22
> > **Thanks for your replies**
> >
> > Thank you for your replies to my questions. I look forward to seeing more images, especially the dynamic visualization you mentioned. I also appreciate the expansion of the paper to other sequence types, though including lots of large tables of results is a poor use of space - much of this could have been added to the appendix, which would have made space for additional exposition of the method.
> >
> > While I think this representation method is interesting by itself, I still assert that for it to be taken seriously, it needs to be benchmarked against SOTA sequence models on these tasks, not just benchmarked against other similar methods. Stating that ProteinBERT didn't work well is a start, but a more systematic comparison is needed to decide whether to use this method at all when approaching other practical problems.
> >
> > In looking over the other reviews, I want to mention that I'm sympathetic to Reviewer B9o8's comments about this not actually being a learning algorithm and therefore not appropriate for ICLR. I thought about this too, but decided to be liberal in my interpretation. I am now wondering if that was the right choice. I will keep my rating as is for now, but would look to other reviewers and the AC to help inform this point.

---

> > > ### Comment · Reviewer_TS9L · 2023-11-22
> > > **Minor stylistic comments**
> > >
> > > - it is SMILES, not SMILE
> > > - It is ResNet-50, not RESNET50

---

> > > > ### Author Response · Authors · 2023-11-22
> > > > **Authors response to minor stylistic comments**
> > > >
> > > > ```
> > > > it is SMILES, not SMILE
> > > > It is ResNet-50, not RESNET50
> > > > ```
> > > >
> > > > We have adjusted the content accordingly in the revised PDF submission.

---

> > > ### Author Response · Authors · 2023-11-22
> > > **Authors Response to Reviewer Further Comments**
> > >
> > > We thank the reviewer for their comments
> > >
> > > ```
> > > Thank you for your replies to my questions. I look forward to seeing more images, especially the dynamic visualization you mentioned. I also appreciate the expansion of the paper to other sequence types, though including lots of large tables of results is a poor use of space - much of this could have been added to the appendix, which would have made space for additional exposition of the method.
> > > ```
> > >
> > > We will adjust the content and move the mentioned stuff to the appendix in the camera-ready version of the paper. For now, our idea is to include new results in the main paper so that they are easily visible to the readers.
> > >
> > > ```
> > > While I think this representation method is interesting by itself, I still assert that for it to be taken seriously, it needs to be benchmarked against SOTA sequence models on these tasks, not just benchmarked against other similar methods. Stating that ProteinBERT didn't work well is a start, but a more systematic comparison is needed to decide whether to use this method at all when approaching other practical problems.
> > > ```
> > >
> > > We partially agree with the reviewer. Please note that the new baseline model that we included in our revision, i.e. Spike2CGR, is a very recent method and was published this year. That method is basically the SOTA in such sequence-to-image transformation domain. Since we beat that method, it means that we beat all methods that Spike2CGR claims to beat, hence we are achieving state-of-the-art results. Moreover, the ProteinBert itself was proposed last year (i.e. in 2022). While it is true that the results will be more solid to include comparisons with other methods, we want to mention that since such a domain is not very new (i.e. traditional sequence-to-image transformation methods like Chaos game representation only work for biological sequences while traditional sequence analysis methods cannot use sophisticated vision models for analysis, and our method is bridging that gap), we would leave such comparisons for the future extensions of this idea. Since we have a competitor from 2023 (i.e. Spike2CGR) and from 2022 (i.e. ProteinBert), we believe that such methods themselves are able to show a comprehensive comparison from literature (among other baselines that we included). Further discussion among researchers and future versions of this idea would definitely help researchers decide whether to use this method at all when approaching other practical problems. For now, since we beat the recently proposed methods in both sequence-to-image transformations (e.g. Spike2CGR) and sequence-only (e.g. ProteinBert) based methods, we would recommend researchers use this method for other practical problems.
> > >
> > > ```
> > > In looking over the other reviews, I want to mention that I'm sympathetic to Reviewer B9o8's comments about this not actually being a learning algorithm and therefore not appropriate for ICLR. I thought about this too, but decided to be liberal in my interpretation. I am now wondering if that was the right choice. I will keep my rating as is for now, but would look to other reviewers and the AC to help inform this point.
> > > ```
> > >
> > > Thanks for sharing your insights. Just for further clarity, our idea is ``creating a representation for learning``.

---

> > > ### Author Response · Authors · 2023-11-22
> > > **Authors response based on new baseline**
> > >
> > > Dear Reviewer, as per your comment regarding the inclusion of more SOTA sequence methods, we have now included a String Kernel-based method as another state-of-the-art method, which is taken from [1] and [2]. Since this method is purely designed for the strings/sequences and the method was originally proposed in NIPS [2] with its extension in TCBB [1], we believe that this could perfectly be justified as the SOTA sequence method. A brief description of the method is now included in Table 2 in the revised PDF. The results are included in Tables 4, 5, 6, and 7 in the revised manuscript.
> > >
> > > We can observe from the results that our sequence-to-image transformation-based method significantly outperforms the string kernel-based method for music and protein subcellular datasets.
> > >
> > > Just for clarification, we have now SOTA methods from NeurIPS (String Kernel) [2] and Machine Learning Journal (Spike2CGR) [3]. Moreover, in the revised manuscript, we show results for datasets from different domains including protein sequences, nucleotide sequences, SMILES strings, and Music sequences. This overall inclusion of new SOTA methods along with results for new datasets are part of our effort to show the generalizability of our method and its applicability in different domains. Hence we believe that the ICLR community can greatly benefit from such an idea and this could initiate great discussions among researchers at the conference.
> > >
> > >
> > >
> > > References:
> > >
> > > [1] Ali S, Sahoo B, Khan MA, Zelikovsky A, Khan IU, Patterson M. Efficient approximate kernel based spike sequence classification. IEEE/ACM Transactions on Computational Biology and Bioinformatics (TCBB). 2022 Sep 14.
> > >
> > > [2] Farhan M, Tariq J, Zaman A, Shabbir M, Khan IU. Efficient approximation algorithms for strings kernel based sequence classification. Advances in neural information processing systems (NIPS). 2017;30.
> > >
> > > [3] Murad T, Ali S, Khan I, Patterson M. Spike2CGR: an efficient method for spike sequence classification using chaos game representation. Machine Learning. 2023 Aug 28:1-26.

---

### Official Review · Reviewer_B9o8 · 2023-10-31

**Soundness:** 3 good
**Presentation:** 2 fair
**Contribution:** 2 fair
**Rating:** 3
**Confidence:** 3

**Summary:**

The paper presents a methodology for encoding protein sequences as images in a 2D plane with the use of Bézier curves. Protein and DNA sequences have been mapped into images in the past to take advantage of machine learning methods, including CNNs, and the proposed approach brings a novel transformation that preserves meaningful biological information.

**Strengths:**

* The paper addresses an important problem, which is the representation of biological sequences.
* The results indicate that the proposed approach has clear advantages over prior methodologies, both quantitatively and qualitatively.
* The benchmarking exercise is extensive. The paper evaluates various architectures and tests several baseline approaches in different datasets.

**Weaknesses:**

* From the machine learning perspective, the paper does not have a significant contribution. While the problem is important and the results are very promising, the technical novelty is limited. Perhaps this piece of work can be better appreciated by the bioinformatics community.
* The proposed approach is not a learning algorithm, where the representation is learned automatically. Instead, the approach is a transformation of individual data points into a different representation, which is shown to be more effective for machine learning algorithms.
* The resulting images are not presented, even though the manuscript claims that these are more interpretable and meaningful. A qualitative comparison of how the images look like with respect to previous attempts for a given sequence (or a portion of it), would be helpful.
* The paper devotes much space in tables with dataset and methods descriptions. This space could be better utilized with different analysis and other results.

**Questions:**

Unfortunately, I don't think the paper is a good fit for this venue. This does not mean that the paper is incorrect or has major mistakes, it is just that the audience may not be the correct community to present and discuss its true value. The idea is great, and I encourage the authors to consider submitting to a bioinformatics journal or similar, where the machine learning contribution is not expected to be the central contribution of the manuscript.

---

> ### Author Response · Authors · 2023-11-20
> **Authors Response to the Reviewers Comments 1/2**
>
> We thank the reviewer for their comments. Please find below our detailed response to each comment
>
> ```
> From the machine learning perspective, the paper does not have a significant contribution. While the problem is important and the results are very promising, the technical novelty is limited. Perhaps this piece of work can be better appreciated by the bioinformatics community.
> ```
>
> We respectfully would like to disagree with the reviewer. The main idea of this paper is to transform any type of sequence into an image-based representation so that we can open the doors of applying vision models for better classification. Since there is more sequence-based data available for the biological sequences, we decided to use different types of such protein sequences to show the performance of the proposed method. However, the prove thethe relevance to ICLR community point, we did the following modifications
>
> To show the generalizability, we now included three more datasets that are non-protein sequences. Their details are shown in Table 1 in the revised paper. Firstly, we use human DNA data, which contains nucleotide sequences. The second data that we use now comprises SMILES strings, which are comprised of drug molecules. The third data that we use is the music sequence data, where the audio sequences belong to 10 unique music genres. Now, using protein sequences (as in the initial submission) along with the nucleotide sequences, SMILES strings, and music sequences datasets helps us to show that the proposed method can generalize on sequences from different domains. Here please note that the majority of the baselines (other than RandomCGR) cannot operate on SMILES and Music data since the number of unique characters is>20 and the baseline methods cannot operate on such complex datasets by design. The results for the human DNA data, SMILES string data, and Music sequence data are shown in Table 5, 6, and 7 in the revised version. Compared to Spike2CGR, we can observe that the proposed method outperforms in the majority of the scenarios. Similarly, on all new datasets, the proposed method performance significantly improves for the pre-trained vision models.
>
> Considering that the ICLR community is interested in the "Learning Representation" work, we are proposing a "representation learning" method that takes sequences as input and transforms it into image-based "Representation", which can then be used for downstream classification task. However, to better show a connection to the machine learning domain, we now included Spike2CGR[1] as one of the baselines (detail included in Table 2 in the revised paper), which works on a similar problem domain for the protein sequences and has recently been published in the "Springer Machine Learning Journal". We believe that considering such a method as a baseline could help to justify that the problem is indeed important for the machine learning community as that work is published in one of the highly rated machine learning journals.
>
> We now believe that considering a recent method as a baseline from ML journal along with the new datasets from non-protein sequence domain could help to show the generalizability of proposed method and relevance to the ICLR community.
>
>
> ```
> The proposed approach is not a learning algorithm, where the representation is learned automatically. Instead, the approach is a transformation of individual data points into a different representation, which is shown to be more effective for machine learning algorithms.
> ```
>
> We would like to clarify that the learning of images from the sequences is done in a completely unsupervised way. We proposed an algorithm that "learns" a representation in the form of the image where we have sequences as an input.
>
> ```
> The resulting images are not presented, even though the manuscript claims that these are more interpretable and meaningful. A qualitative comparison of how the images look like with respect to previous attempts for a given sequence (or a portion of it), would be helpful.
> ```
>
> We already presented the images for the Bezier curve in Figure 5 in the appendix. Moreover, we provide more images for the baselines as well in Figure 7 in the appendix. We would request the reviewer to please check the images along with further results and discussion mentioned in the appendix. We did not mention such an analysis in the main paper due to space limitations.

---

> > ### Comment · Reviewer_B9o8 · 2023-11-21
> > **Thanks for the comments + more questions**
> >
> > I want to thank the authors for addressing my concerns and for adding more experiments and results to the paper. It is indeed great to see that the method can work with other types of sequences. I still have a few more questions to make sure I understand the responses and the claims of the paper:
> >
> > 1. Is reusing vision models to improve classification performance the main motivation to transform sequences into images? I find the reasoning a bit strange, as the goal of processing sequences should be to do better computation and better understand the sequences.
> > 2. Have you considered comparing against other sequence (not image-based) models to solve these tasks?
> > 3. Is the transformation from sequence to image invertible or reversible?
> > 4. What is the learning mechanism of the proposed approach? Maybe I missed this, but I thought the transform was based on single data points rather than learned from a dataset and transferred to new data points. The optimization in Algorithm 1 and Figure 1 seems to depend on a single sequence rather than a training set, which makes me think that there is no training (no loss function), and therefore, no learning. I agree that the there is a representation change (from sequence to image) but I don't see it being learned.
> >
> > I appreciate if the authors can clarify these additional questions.
> > Thank you!

---

> > > ### Author Response · Authors · 2023-11-21
> > > **Authors response to reviewer's more comments**
> > >
> > > We thank the reviewer for acknowledging our efforts of adding new experiments in the revised paper and addressing major concerns. Please find below our response to the new comments.
> > >
> > > ```
> > > Is reusing vision models to improve classification performance the main motivation to transform sequences into images? I find the reasoning a bit strange, as the goal of processing sequences should be to do better computation and better understand the sequences.
> > > ```
> > >
> > > We want to clarify that the ultimate goal is to perform sequence processing and a better understanding of the sequences. In doing so, there are two methods. The first one deals with designing embeddings (vector representations) directly from the sequences and using classical ML and DL models to perform classification. However, we find out that such a type of embedding-based (tabular) representation shows promising results using classical ML models, but DL models do not perform state-of-the-art on such tabular data [1] [2]. Therefore, we thought of using vision models as an alternative to the DL models that take vector-based representations as input. Here comes the second type of method, where we transform the sequences into images and then see how much better performance (compared to traditional embedding methods) we can achieve. Therefore, we decided to go with designing such a type of transformation method.
> > >
> > > ```
> > > Have you considered comparing against other sequence (not image-based) models to solve these tasks?
> > > ```
> > >
> > > Yes, we already showed results for non-image-based representation learning models. Those methods are mentioned in Table 2 of the main paper. Particularly, we used one-hot encoding (OHE) and Wasserstein distance-guided representation (WDGRL) [3] to generate the embeddings, which are then used as input to perform classification. Their results are shown in Tables 4, 5, 6, and 7.  We also tried methods like ProteinBert [4] but their results were also not promising, hence not included in the paper.
> > >
> > > ```
> > > Is the transformation from sequence to image invertible or reversible?
> > > ```
> > >
> > > The transformation is not reversible. Our goal is not to generate a reversible algorithm in this case, hence we only focused on generating images from sequences, not the other way around.
> > >
> > > ```
> > > What is the learning mechanism of the proposed approach? Maybe I missed this, but I thought the transform was based on single data points rather than learned from a dataset and transferred to new data points. The optimization in Algorithm 1 and Figure 1 seems to depend on a single sequence rather than a training set, which makes me think that there is no training (no loss function), and therefore, no learning. I agree that the there is a representation change (from sequence to image) but I don't see it being learned.
> > > ```
> > >
> > > We want to mention that there is no loss function type logic in our algorithm. By term learning, we mean that the final type of image generated depends on the characters (e.g. amino acids) within any given sequence. A sequence goes as input to our black box (Bezier Curve) model, where our algorithm (inside the logic of the black box) generates coordinate (pixel) values for each of the sequence characters. Then all coordinate values are combined to form an image for any given sequence. We defined every step of our model with a sample running example in Figure 1 as well in the manuscript. Just for clarity purposes, we want to mention that our model does not perform a type of "learning" where one learned model can be transferred to new data points. Rather, our logic/algorithm could be transferred, which works in an unsupervised manner to generate images from any type of input sequences.
> > >
> > > We again want to thank you for your valuable feedback. If you have any other questions or need further clarification, please do let us know. If you are satisfied with the overall idea of our paper and our response so far, we would request you to kindly consider increasing your score (if possible), so that we may have better chances of acceptance. We believe that if accepted, such type of idea can create greater interests (and discussions) among researchers in the ICLR community.
> > >
> > > References:
> > >
> > > [1] Tabular Data: Deep Learning is Not All You Need (https://arxiv.org/abs/2106.03253)
> > >
> > > [2] Why do tree-based models still outperform deep learning on tabular data? (https://arxiv.org/abs/2207.08815)
> > >
> > > [3] Jian Shen, Yanru Qu, Weinan Zhang, and Yong Yu. Wasserstein distance guided representation learning for domain adaptation. In AAAI conference on artificial intelligence, 2018.
> > >
> > > [4] Brandes N, Ofer D, Peleg Y, Rappoport N, Linial M. ProteinBERT: a universal deep-learning model of protein sequence and function. Bioinformatics. 2022 Apr 15;38(8):2102-10.

---

> > > ### Author Response · Authors · 2023-11-22
> > > **Authors further response to reviewer comment on "not image-based" models**
> > >
> > > Dear Reviewer, as per your comment regarding the inclusion of "not image-based" models, we have now included a String Kernel-based method as another state-of-the-art method, which is taken from [1] and [2]. Since this method is purely designed for the strings/sequences and the method was originally proposed in NIPS [2] with its extension in TCBB [1], we believe that this could perfectly be justified as the SOTA sequence method. A brief description of the method is now included in Table 2 in the revised PDF. The results are included in Tables 4, 5, 6, and 7 in the revised manuscript.
> > >
> > > We can observe from the results that our sequence-to-image transformation-based method significantly outperforms the string kernel-based method for music and protein subcellular datasets.
> > >
> > > Just for clarification, we have now SOTA methods from NeurIPS (String Kernel) [2] and Machine Learning Journal (Spike2CGR) [3]. Moreover, in the revised manuscript, we show results for datasets from different domains including protein sequences, nucleotide sequences, SMILES strings, and Music sequences. This overall inclusion of new SOTA methods along with results for new datasets are part of our effort to show the generalizability of our method and its applicability in different domains. Hence we believe that the ICLR community can greatly benefit from such an idea and this could initiate great discussions among researchers at the conference.
> > >
> > >
> > >
> > > References:
> > >
> > > [1] Ali S, Sahoo B, Khan MA, Zelikovsky A, Khan IU, Patterson M. Efficient approximate kernel based spike sequence classification. IEEE/ACM Transactions on Computational Biology and Bioinformatics (TCBB). 2022 Sep 14.
> > >
> > > [2] Farhan M, Tariq J, Zaman A, Shabbir M, Khan IU. Efficient approximation algorithms for strings kernel based sequence classification. Advances in neural information processing systems (NIPS). 2017;30.
> > >
> > > [3] Murad T, Ali S, Khan I, Patterson M. Spike2CGR: an efficient method for spike sequence classification using chaos game representation. Machine Learning. 2023 Aug 28:1-26.

---

> ### Author Response · Authors · 2023-11-20
> **Authors Response to the Reviewers Comments 2/2**
>
> ```
> The paper devotes much space in tables with dataset and methods descriptions. This space could be better utilized with different analysis and other results.
> ```
>
> The reviewer rightfully pointed out that some tables can go to the appendix and space could be better utilized with different analyses. In our initial submission, we tried to include important information in the main paper and moved the remaining content to the appendix. However, since we now included new results for the three new datasets along with further discussion based on the reviewer's comments, we will adjust the content as per instructions in the final camera-ready version.
>
> ```
> Unfortunately, I don't think the paper is a good fit for this venue. This does not mean that the paper is incorrect or has major mistakes, it is just that the audience may not be the correct community to present and discuss its true value. The idea is great, and I encourage the authors to consider submitting to a bioinformatics journal or similar, where the machine learning contribution is not expected to be the central contribution of the manuscript.
> ```
> We thank the reviewer for appreciating the quality of our work. As we mentioned in our detailed response, we believe that we proposed a "representation learning" idea that can take any type of sequence-based data as input e.g. protein sequences, nucleotide sequences, SMILES strings, and Music sequences. Moreover, we use a recently proposed method from ML journal as a baseline to show relevance to the machine learning community along with the "representation learning" community. We hope that our argument will be enough to convince the reviewer to reconsider their decision. If there any any other points that the reviewer would like to mention, we would be very interested in considering those.
>
>
> References:
>
> [1] Murad T, Ali S, Khan I, Patterson M. Spike2CGR: an efficient method for spike sequence classification using chaos game representation. Machine Learning. 2023 Aug 28:1-26.

---

> ### Author Response · Authors · 2023-11-21
> **Second Response To The Reviewer's Comments**
>
> Dear reviewer, as the end of the discussion period is fast approaching, please do let us know if you have any further suggestions or concerns to which we can respond. We'd be interested to hear if our response and changes to the manuscript helped answer your questions. If you are satisfied with our response, we would kindly request you to increase your score if possible, so that we have more chances of paper acceptance.

---

### Official Review · Reviewer_Bdp5 · 2023-11-01

**Soundness:** 4 excellent
**Presentation:** 3 good
**Contribution:** 4 excellent
**Rating:** 6
**Confidence:** 3

**Summary:**

The authors propose an approach to transform biological sequences into images using Bezier curves for element mapping. A core motivation is enhancing the representation of sequence information in the generated images, as traditional methods like CGR tends to map elements to a limited set of pixels. Through experiments on three protein sequence datasets for tasks including subcellular localization and host prediction, the authors demonstrate the superiority of the proposed Bezier curve encoding method, with significant gains in accuracy over baselines.

Additionally, the smooth interpolation of control points enabled by Bezier curves is cited as improving interpretability of the visualizations. Overall, encoding sequences as Bezier curve images appears promising as it provides richer representations that translate to markedly improved performance on downstream classification tasks.

**Strengths:**

- The proposed sequence-to-image transformation technique is novel and a key contribution that could prove widely applicable and beneficial in the field. The simple and straightforward image generation process based on standard Bezier curve equations is clearly explained.
- The authors introduce careful methodological choices; e.g., introducing controlled randomness via deviations to reveal hidden patterns, to overcome limitations of fixed mappings in CGR.
- The authors demonstrated the usefulness of their information-rich curve-based image representations by validating on multiple protein sequence datasets for subcellular localization task. The proposed approach substantially improved classification performance over baseline CGR, highlighting the benefits of the Bezier representation. In addition, the authors also explored the potential of the smoother Bezier curve interpolations in improving interpretability compared to sparse CGR images.
-encoding method is agnostic to the choice of downstream classifier, allowing flexible integration with existing pipelines.
- The proposed approach could generalize well to other types of biological sequences like DNA beyond tested protein use cases.

**Weaknesses:**

- The proposed representations might not encode signal about sequential information as amino acid order is not explicitly encoded, which may prove to be necessary in some usecases.
- Limited ablation studies were performed to analyze the impact of key parameters like number of control points and deviations.
- (minor) There is a notable computational overhead to the proposed approach, although at a benefit of improved performance.

**Questions:**

- How does the performance scale with much longer input sequences? At what sequence length does the image encoding become unwieldy?
Opensourcing the code will help reproducibility efforts and also drive adoption of this approach.

---

> ### Author Response · Authors · 2023-11-20
> **Authors Response to the Reviewers Comments**
>
> We would like to thank the reviewer for appreciating our work and giving us a valuable response. Please find below our response to the comments.
>
> ```
> The proposed representations might not encode signal about sequential information as amino acid order is not explicitly encoded, which may prove to be necessary in some usecases.
> ```
>
> The reviewer rightfully pointed out that order information could be important in some cases, e.g. virus classification where there are virus and non-virus mutations in the biological sequences. However, we want to mention that our the specific scenario, the order of the sequence is not important. The order information comes into play when we want to design numerical vector-based representation. For the image transformation, we are only interested in capturing the mutations, which are a kind of differentiating factor among different class images.
>
> ```
> Limited ablation studies were performed to analyze the impact of key parameters like number of control points and deviations.
> ```
>
> We will add detailed plots to show the impact of the mentioned parameter on predictive performance in the camera-ready version of the paper.
>
>
> ```
> (minor) There is a notable computational overhead to the proposed approach, although at a benefit of improved performance.
> ```
>
> We thank the reviewer for pointing this out. We want to mention that the proposed method is not parameter-heavy. Since we only use two parameters, i.e.  $m$ (the number of points along each curve) and $ite$ (line 10 in Algorithm 1), which is the number of deviation pair points, the image generation using the proposed method is very fast.
>
> ```
> How does the performance scale with much longer input sequences? At what sequence length does the image encoding become unwieldy? Open sourcing the code will help reproducibility efforts and also drive the adoption of this approach.
> ```
>
> Since the proposed method does not have a lot of parameters, having longer-length sequences does not affect scalability performance much. The maximum length for any sequence we tried is for the Human DNA dataset (included in Table 1 of the revised version), where the max length is $18921$. On such bigger length sequences, the image generation time was not affected much. We also want to mention that we do plan to make our code open-source. The main reason for not doing it as part of submission is to maintain anonymity. In the final camera-ready version of the paper, we will provide a URL to the GitHub repository containing our code.

---

> > ### Author Response · Authors · 2023-11-21
> > **Second Response To The Reviewer's Comments**
> >
> > Dear reviewer, as the end of the discussion period is fast approaching, please do let us know if you have any further suggestions or concerns to which we can respond. We'd be interested to hear if our response and changes to the manuscript helped answer your questions.

---

### Official Review · Reviewer_FrGV · 2023-11-04

**Soundness:** 3 good
**Presentation:** 3 good
**Contribution:** 3 good
**Rating:** 5
**Confidence:** 2

**Summary:**

The paper proposes a new method to transform biological sequences, like protein or DNA sequences, into images using Bezier curves. This allows applying deep learning image models to analyze the sequences. The authors mention that existing/traditional methods convert sequences into numerical features or use Chaos Game Representation (CGR) to create images. But these have limitations like sparsity, high dimensionality, poor representation of sequence information in images.

The proposed method maps each element (amino acid, nucleotide) of a sequence onto a Bezier curve to create an image. Multiple points on the curve represent each element. This captures more information and patterns in the image compared to traditional CGR-based methods where elements map to fixed pixels. Experiments on three protein datasets for classification tasks show the Bezier method outperforms baselines like FCGR and RandomCGR images, and numerical embedding methods.

In the results, the authors demonstrate that for subcellular protein localization dataset, Bezier images achieve 40% higher accuracy than FCGR using CNNs. Furthermore, cluster visualization and histograms also show Bezier embeddings preserve structure better. Overall, the paper demonstrates biological sequence analysis benefits from transforming sequences into images via Bezier curves before applying deep learning models.

**Strengths:**

+ The paper describes a new idea of using Bezier curves to create visual representations of biological sequences.
+ The results also demonstrate improved performance over baseline methods on several protein sequence classification tasks.

**Weaknesses:**

- In its current format, the paper may not be a good fit for ICLR audience, and the featurization is specific to a small domain of problem.
- The demonstrated evaluation is limited to only protein sequence datasets related to subcellular localization, virus hosts, etc. It was not clear to me how this idea generalizes to broader range of biological problems.

**Questions:**

1) How much hyper-parameter optimization is needed for this approach? Are these B'ezier features easy to generate
2) Can this idea be generalized to other areas beyond biological approach? Please comment on the generality of the featurization and fit with ICLR audience.

---

> ### Author Response · Authors · 2023-11-20
> **Authors Response to the Reviewers Comments**
>
> We thank the reviewer for their comments. Below, please find our detailed response to each of the comment.
>
> ```
> In its current format, the paper may not be a good fit for ICLR audience, and the featurization is specific to a small domain of problem.
> ```
>
> We respectfully would like to disagree with the reviewer. The main idea of this paper is to transform any type of sequence into an image-based representation so that we can open the doors of applying vision models for better classification. Since there is more sequence-based data available for the biological sequences, we decided to use different types of such protein sequences to show the performance of the proposed method. However, the prove the generalizability and the relevance to ICLR community point, we did the following two modifications
>
> ``Generalizability``: To show the generalizability, we now included three more datasets that are non-protein sequences. Their details are shown in Table 1 in the revised paper. Firstly, we use human DNA data, which contains nucleotide sequences. The second data that we use now comprises SMILES strings, which are comprised of drug molecules. The third data that we use is the music sequence data, where the audio sequences belong to 10 unique music genres. Now, using protein sequences (as in the initial submission) along with the nucleotide sequences, SMILES strings, and music sequences datasets helps us to show that the proposed method can generalize on sequences from different domains. Here please note that the majority of the baselines (other than RandomCGR) cannot operate on SMILES and Music data since the number of unique characters is>20 and the baseline methods cannot operate on such complex datasets by design. The results for the human DNA data, SMILES string data, and Music sequence data are shown in Table 5, 6, and 7 in the revised version. Compared to Spike2CGR, we can observe that the proposed method outperforms in the majority of the scenarios. Similarly, on all new datasets, the proposed method performance significantly improves for the pre-trained vision models.
>
> ``Relevance to ICLR community:`` Considering that the community is interested in the "Learning Representation" work, we are proposing a "representation learning" method that takes sequences as input and transforms it into image-based "Representation", which can then be used for downstream classification task. However, to better show a connection to the machine learning domain, we now included Spike2CGR[1] as one of the baselines (detail included in Table 2 in the revised paper), which works on a similar problem domain for the protein sequences and has recently been published in the "Springer Machine Learning Journal". We believe that considering such a method as a baseline could help to justify that the problem is indeed important for the machine learning community as that work is published in one of the highly rated machine learning journals.
>
> We now believe that considering a recent method as baseline from ML journal along with the new datasets from non-protein sequence domain could help to show the generalizability of proposed method and relevance to the ICLR community.
>
> ```
> The demonstrated evaluation is limited to only protein sequence datasets related to subcellular localization, virus hosts, etc. It was not clear to me how this idea generalizes to broader range of biological problems.
> ```
> As described above, we now expanded our experimental setting by using broader range of biological datasets including nucleotide and SMILES string based sequences as well as Music sequences datasets.
>
> ```
> How much hyper-parameter optimization is needed for this approach? Are these B'ezier features easy to generate
> ```
> For the Bézier Curve Based Image Generation, there are only two parameters. The first parameter is $m$, which is the number of points along each curve. The second parameter is $ite$ (line 10 in Algorithm 1), which is the number of deviation pair points. Hence our model is not parameter-heavy. However, please note that the hyperparameters for the vision models is different from the Bézier Curve model as the vision models are only used for the downstream classification task. The details regarding hyperparameters of vision models is already given in Section 4.3.1.
>
> ```
> Can this idea be generalized to other areas beyond biological approach? Please comment on the generality of the featurization and fit with ICLR audience.
> ```
> The proposed method can work on any type of sequence data. We already demonstrated that in our above response and in the revised submission, where we included results for nucleotides, SMILES, and Music-based sequence datasets.
>
>
> References:
>
> [1] Murad T, Ali S, Khan I, Patterson M. Spike2CGR: an efficient method for spike sequence classification using chaos game representation. Machine Learning. 2023 Aug 28:1-26.

---

> > ### Author Response · Authors · 2023-11-21
> > **Second Response To The Reviewer's Comments**
> >
> > Dear reviewer, as the end of the discussion period is fast approaching, please do let us know if you have any further suggestions or concerns to which we can respond. We'd be interested to hear if our response and changes to the manuscript helped answer your questions. If you are satisfied with our response, we would kindly request you to increase your score if possible, so that we have more chances of paper acceptance.

---

### Author Response · Authors · 2023-11-20
**General Response to All reviewers and further clarifications**

Dear reviewers, program chairs, and area chairs,

We would like to thank everyone for their valuable comments. Here, we want to give overall clarifications based on all reviewer's comments. We noticed that some concerns were raised regarding the suitability of the venue for our paper since the paper seems to focus more on the healthcare domain. However, we argue that our "representation learning" method, where we propose an algorithm to transform sequences into images, is a general-purpose method and can work for sequences beyond the protein sequences. To prove this, we showed (in the revised version) experiments on nucleotide sequences, SMILES strings, and Music sequence datasets. Moreover, we included a baseline from the ML journal to show the paper's relevance to the ML community.

Also, we now removed the word "biological" from the title of our paper (in the revised version of the PDF). We also carefully used this word so that the paper doesn't make a perception of being only interesting to the biology domain.

Moreover, since our model is not too hyperparameter-heavy, we believe that the readers from the ICLR community would be interested in exploring such ideas for diverse types of sequence analysis in the future, for which, our paper can act as a benchmark. Since this type of domain is comparatively new, we believe that our work can open up new discussions regarding how to deal with non-image-based datasets using state-of-the-art vision models, which would be an interesting topic among researchers from the ICLR community.

---

> ### Author Response · Authors · 2023-11-22
> **Extended General Response to All reviewers regarding the inclusion of non-image-based SOTA method**
>
> Dear reviewers, program chairs, and area chairs,
>
> We saw a few comments regarding the inclusion and discussion of non-image-based methods for comparison with the proposed method. In that effort, we have now included a String Kernel-based method as another state-of-the-art method, which is taken from [1] and [2]. Since this method is purely designed for the strings/sequences and the method was originally proposed in NIPS [2] with its extension in TCBB [1], we believe that this could perfectly be justified as the SOTA sequence method. A brief description of the method is now included in Table 2 in the revised PDF. The results are included in Tables 4, 5, 6, and 7 in the revised manuscript.
>
> We can observe from the results that our sequence-to-image transformation-based method significantly outperforms the string kernel-based method for music and protein subcellular datasets.
>
> Just for clarification, we have now SOTA methods from NeurIPS (String Kernel) [2] and Machine Learning Journal (Spike2CGR) [3]. Moreover, in the revised manuscript, we show results for datasets from different domains including protein sequences, nucleotide sequences, SMILES strings, and Music sequences. This overall inclusion of new SOTA methods along with results for new datasets are part of our effort to show the generalizability of our method and its applicability in different domains. Hence we believe that the ICLR community can greatly benefit from such an idea and this could initiate great discussions among researchers at the conference.
>
>
>
> References:
>
> [1] Ali S, Sahoo B, Khan MA, Zelikovsky A, Khan IU, Patterson M. Efficient approximate kernel based spike sequence classification. IEEE/ACM Transactions on Computational Biology and Bioinformatics (TCBB). 2022 Sep 14.
>
> [2] Farhan M, Tariq J, Zaman A, Shabbir M, Khan IU. Efficient approximation algorithms for strings kernel based sequence classification. Advances in neural information processing systems (NIPS). 2017;30.
>
> [3] Murad T, Ali S, Khan I, Patterson M. Spike2CGR: an efficient method for spike sequence classification using chaos game representation. Machine Learning. 2023 Aug 28:1-26.

---

> > ### Author Response · Authors · 2023-11-23
> > **Interactive Plot For Better Understanding of our idea**
> >
> > Dear reviewers, program chairs, and area chairs,
> >
> > To show the interactive plot demo, we have added a sample Jupyter notebook code into the supplementary material (i.e. Bezier_Interactive_Plot jupyter notebook file), which, if you run, will generate a kind of interactive plot where you can visually see how the bezier curve-based plot is generated for each amino acid within a randomly generated protein sequence.
> >
> > Due to the annonimity, we show the interactive plot for a sample sequence rather than actual data as it is easy to run just to get an idea of how our method works. We hope that this will help you understand our idea better. In the final camera-ready version of the paper, we will include such interactive plot in the publicly available code so that readers can use that to understand our idea better.

---

### Meta-Review · Area_Chair_ufVq · 2023-12-08

**Metareview:**

This study introduces a method for converting biological sequences (eg. DNA) into images using Bézier curves. Traditional methods of sequence analysis often convert these sequences into numerical data or use Chaos Game Representation (CGR). The proposed Bézier curve approach addresses these issues by mapping each element of a sequence onto a curve.

The paper reports that this method enhances the quality of the data for CNNs. Experiments on three different protein sequence datasets show that the method consistently outperformed more traditional methods (eg. FCGR, RandomCGR) -- in tasks such as protein subcellular location prediction and coronavirus host classification, the Bézier curve images achieved significantly higher accuracy.

**Justification For Why Not Higher Score:**

This submission was considered borderline. There was significant interaction between the authors and the reviewers. It is evident that further refinement of the paper could enhance its appeal to a broader audience. For the paper's improvement, it would be advisable to resubmit a revised version, potentially to a different publication venue.

**Justification For Why Not Lower Score:**

N/A.

---

### Decision · Program_Chairs · 2024-01-16

Reject